# Novel metrics based on Biogeochemical Argo data to improve the model uncertainty evaluation of the CMEMS Mediterranean marine ecosystem forecasts

Stefano Salon[1], Gianpiero Cossarini[1], Giorgio Bolzon[1], Laura Feudale[1], Paolo Lazzari[1], Anna Teruzzi[1], Cosimo Solidoro[1], Alessandro Crise[1]

[1]Oceanography Section, OGS - Istituto Nazionale di Oceanografia e di Geofisica Sperimentale, Trieste, 34151, Italy

*Correspondence to*: Stefano Salon (ssalon@inogs.it)

**Abstract.** The quality of the upgraded version of the CMEMS biogeochemical operational system of the Mediterranean Sea (MedBFM) is assessed in terms of consistency and forecast skill, following a mixed validation protocol that exploits different reference data from satellite, oceanographic databases, Biogeochemical Argo floats and literature. We show that the quality of the MedBFM system has been improved in the previous 10 years. We demonstrate that a set of metrics based on the GODAE paradigm can be efficiently applied to validate an operational model system for biogeochemical and ecosystem forecasts. The accuracy of the CMEMS biogeochemical products for the Mediterranean Sea can be achieved from basin-wide and seasonal scale to mesoscale and weekly scale, and its level depends on the specific variable and the availability of reference data, the latter being an important prerequisite to build robust statistics. In particular, the use of the Biogeochemical Argo floats data proved to significantly enhance the validation framework of operational biogeochemical models. New skill metrics, aimed to assess key biogeochemical processes and dynamics (e.g. deep chlorophyll maximum depth, nitracline depth), can be easily implemented to routinely monitor the quality of the products and highlight possible anomalies through the comparison of NRT forecasts skill with pre-operationally defined seasonal benchmarks. Feedbacks to the observing autonomous systems in terms of QC and deployment strategy are also discussed.

## 1 Introduction

Operational ocean forecasting systems integrate remote observations, in situ measurements and modelling systems, and have been widely recognized as important assets for ocean state monitoring (von Schuckmann et al., 2016) and the development of the blue economy (She et al., 2016). In such framework, the operational monitoring and forecasting of marine biogeochemistry and ecosystem dynamics is based on biogeochemical models designed to represent the lower trophic level ecosystem (i.e. from phytoplankton to zooplankton). The improvement of their predictive capability on weekly and seasonal time-scales mostly required by users is strongly related to the development of data assimilation capacity, while their quality assessment is constrained by the availability of reference data, both remote and in situ (Gehlen et al., 2015), and possibly independent (i.e., not assimilated; Gregg et al., 2009). In this perspective, efforts to establish a stronger link between operational biogeochemistry

products and potential users from the fisheries and environmental science communities are constantly increasing (Berx et al., 2011; Payne et al., 2017).

The European Copernicus Marine Environment Monitoring Service (CMEMS; marine.copernicus.eu) operationally provides "regular and systematic core reference information" on the state, variability and dynamics of the ocean, marine ecosystems and sea ice for the global ocean and the European regional seas (Le Traon et al., 2017). As a user-driven service based on a "continuous improvement" philosophy, CMEMS is committed to maintain its operational systems up-to-date in order to supply quality-assessed products for the analysis of the current state of oceans and seas, for the short-term forecasts and for the reanalysis and reprocessing of the recent decades. The CMEMS products are delivered to users through a service portfolio, where the information is organized by data origin, that is from model or from observations (satellite and in situ). Model data from analysis/forecast and reanalyses are geographically grouped for the global ocean and for six regional European regions, with a total of seven specific model systems. Each model system features a physical, a biogeochemical and a wave component. The Mediterranean Sea Monitoring and Forecasting Centre (Med-MFC; Tonani et al., 2013) is one of the regional systems and is composed by the physical system "Med-PHY" (Tonani et al., 2008; Oddo et al. 2009), which drives the biogeochemical system "Med-BIO" (Lazzari et al., 2010, 2012) and the wave system "Med-WAV" (Ravdas et al., 2018). In the recent years, following the CMEMS requirements, the Med-MFC has been consistently upgraded in the physical (Oddo et al. 2014; Clementi et al., 2017; Pistoia et al., 2018), wave (Zacharioudaki et al. 2018) and biogeochemical components (Cossarini et al., 2015; Lazzari et al., 2016), including also data assimilation (Teruzzi et al., 2014; Storto et al., 2015).

More specifically, the last major upgrade of Med-BIO focused on the increase of horizontal resolution from 1/16 to 1/24 degree. The upgrade also involved different aspects of the forecasting system aimed to improve the alignment with the physical component: the new non-linear free-surface curvilinear z*-coordinate configuration used in NEMO3.6 (see Madec, 2016, for the NEMO implementation and further details) and the terrestrial input boundary conditions layout, now including 39 rivers. Moreover, Med-BIO improved the former data assimilation scheme (Teruzzi et al., 2014), extending the assimilation of surface chlorophyll concentration to coastal areas (Teruzzi et al., 2018) and reducing the time-to-solution through a parallelization of the cost function solver (Teruzzi et al., 2019a).

Both historical and near-real time data of observation systems are strategical to evaluate the quality of operational oceanography products (She et al., 2016). However, although operational ocean models are designed to span the whole water column from the surface to the bottom and are now reaching the sub-mesoscale description, deeper ocean and mesoscale remain still not adequately sampled by operational observation systems (Bell et al., 2015; Hernandez et al., 2018). The assessment of the operational ocean products accuracy already benefits from international intercomparison initiatives (e.g., the GOV Task Team for Intercomparison and Validation, ICV-TT; Bell et al., 2015), which also define specific protocols to quantify the quality level of core variables delivered to users (Hernandez et al., 2015; Ryan et al., 2015). This is applicable to the Mediterranean Sea operational systems which include, besides CMEMS Med-MFC, also the Poseidon operational system built on the HYBRID-POM-ERSEM model coupling (Tsiaras et al., 2017; Petihakis et al., 2018).

More in general and concerning biogeochemical applications in the Mediterranean Sea, the limited availability of observational reference data often hinders the validation assessment of model products. The most common approach is based on contrasting model outputs with satellite-derived surface chlorophyll (Tsiaras et al., 2017, for year 2000; Mattia et al., 2013, Macias et al., 2014, Guyennon et al., 2015, Richon et al., 2017, for a portion or the whole investigated multi-year periods). In situ measurements from vessels and scientific cruises are also used in Richon et al. (2017) and Guyennon et al. (2015), but allow only to validate limited temporal and spatial subsets of the simulations (i.e., time series of fixed stations or single transects in a very confined time range). On the other hand, a few basin-wide validation frameworks, especially for nutrients, are based on comparison with climatology, e.g. Tsiaras et al. (2017) used a seasonally aggregated reference for the whole Mediterranean Sea built on 1990-1999 data from SeaDataNet. Generally, modelled vertical properties of biogeochemistry are rarely assessed (e.g. Guyennon et al., 2015 and Teruzzi et al., 2014) due to the lack of adequate reference datasets. In the recent years, the availability of biogeochemical vertical profiles in the Mediterranean Sea has significantly increased with the deployment of Biogeochemical Argo floats (hereafter BGC-Argo floats; Johnson and Claustre, 2016), whose datasets constitute an unprecedented source of reference for biogeochemical model skill assessment, spanning from basin-wide and seasonal scale to mesoscale and weekly scale.

In the present paper, we focus on the CMEMS Mediterranean biogeochemical analysis and forecast system products (delivered from April 2018) and we introduce novel skill metrics based on the comparison between model products and the BGC-Argo floats data. According to the definition adopted within Copernicus community (Hernandez et al., 2018), our model validation follows two main tasks:

1. The pre-operational qualification, that is performed when a new version of the system is developed and a full range of validation metrics is applied to provide an evaluation of the skill performance of the model. The qualification is carried out over a short reanalysis run (e.g. a couple of years) which then provides the initial conditions for the operational analysis and forecast run.

2. The routine, near-real time (NRT) validation of forecast products, that is performed operationally based on the available NRT observations and provides an evaluation of the skill performance of the analysis and forecast products.

The paper is organized as follows. In Section 2 we present the MedBFM system, that is the core of the Med-BIO operational workflow, followed by the reference observations including the recently available BGC-Argo floats data (Section 3). In Section 4 the validation framework is presented, while the most relevant results of the pre-operational and the NRT quality assessment are shown in Section 5. Discussion and conclusions are drawn, respectively, in Sections 6 and 7.

## 2 The Mediterranean Sea Biogeochemical analysis and forecast system

### 2.1 MedBFMv2.1 model system

The Med-BIO analysis and forecast products are provided by the MedBFMv2.1 model system (Fig. 1), which consists of the coupled physical-biogeochemical OGSTM-BFM model and the 3DVarBio assimilation scheme. OGSTM-BFM (Lazzari et

al., 2010, 2012, 2016; Cossarini et al., 2015, and references thereby) is designed with the OGSTM transport model, based on the OPA 8.1 system (Foujols et al., 2000) and a biogeochemical reactor featuring the Biogeochemical Flux Model (BFM; Vichi et al., 2007a,b), which describes the biogeochemical cycles of carbon and macro-nutrients (nitrogen, phosphorus, and silicon) in terms of dynamical interactions among the dissolved inorganic, living organic and non-living organic compartments.

The model presently includes nine plankton functional types (PFTs): phytoplankton PFTs are diatoms, flagellates, picophytoplankton and dinoflagellates; heterotrophic PFTs consist of carnivorous and omnivorous mesozooplankton, bacteria, heterotrophic nanoflagellates and microzooplankton. The non-living compartment consists of three groups: labile, semi-labile and refractory organic matter. The BFM model is also coupled to a carbonate system model (Cossarini et al., 2015, Melaku Canu et al., 2015), which consists of two prognostic state variables: alkalinity (ALK) and dissolved inorganic carbon (DIC)

and provides pH, partial pressure of $CO_2$ (pCO2) and air-sea $CO_2$ flux.

3DVarBio is the variational data assimilation scheme for the update of the four phytoplankton PFTs of BFM using surface chlorophyll retrieved from satellite observations provided by the Ocean Colour Thematic Assembly Centre (OC-TAC) of CMEMS. The 3DVarBio scheme (see details in Teruzzi et al., 2014) decomposes the background error covariance matrix using a sequence of different operators that account separately for the vertical covariance ($V_V$), the horizontal covariance ($V_H$)

and the covariance among biogeochemical variables ($V_b$). $V_V$ is defined by a set of synthetic profiles that are evaluated by means of an Empirical Orthogonal Function (EOF) decomposition applied to a validated multi-annual run (over the period 1998-2015). EOFs are computed for 12 months and 30 coastal and open sea sub-regions in order to account for the variability of 3D chlorophyll anomaly fields. Surface chlorophyll is assimilated over the whole domain, including the coastal areas (Teruzzi et al., 2018), through the upgrade of the non-homogeneous $V_V$ and the non-uniform and direction-dependent $V_H$

specifically focussed for the case 2 waters. Further, the time-to-solution of 3DVarBio has been significantly reduced using the domain decomposition with message passing paradigm to parallelize the code and maximize performance and scalability, and adopting the efficient parallel solver of the PETSc/TAO library for optimizing the cost function minimization (Teruzzi et al., 2019a).

The MedBFMv2.1 system works on a geographical domain that spans from 9°W to 36°E and from 30°N to 46°N with a

meshgrid based on 1/24° longitudinal scale factor and on 1/24°cos(φ) latitudinal scale factor. The vertical meshgrid accounts for 141 vertical z-levels: 35 in the first 200 m depth, 60 between 200 and 2000 m, 28 between 2000 and 4000 m and 18 below 4000. MedBFMv2.1 features the non-linear free surface formulation (Madec et al., 2016) and includes the terrestrial inputs (e.g. nutrients, carbon and alkalinity) from 39 rivers (same as the Med-PHY, see Sect. 2.2 for details) and the Dardanelles treated as a river (Fig. 2).

The MedBFM system is coupled off-line with the Med-PHY system, that provides daily 3D fields of horizontal and vertical current velocity, potential temperature, salinity, vertical eddy diffusivity, and the 2D field of sea surface height (SSH) as forcings for the OGSTM-BFM model. In particular, SSH is used in the new curvilinear z*-coordinate formulation of the MedBFM to compute the vertical scale factor which takes in account the variability of the water column volume, where the vertical coordinate follows the time-dependent non-linear variation of SSH (see Salon et al., 2018). Additional 2D fields from

Med-PHY include the surface data for solar shortwave irradiance and wind stress (derived by the ECMWF atmospheric forcing, see details below), which are used, respectively, as input for the BFM optical module and to solve the air-sea gas exchanges.

The Med-PHY hydrodynamics is solved by the NEMO model (v3.6; Madec et al., 2016) coupled with WaveWatch-III for the wave component and driven by atmospheric forcing of momentum, water and heat fluxes extracted by the 6-hours, 1/8 degree ECMWF operational analysis and forecast fields, plus the daily averaged precipitation and the model predicted surface temperatures (Tonani et al., 2008). The assimilation of in situ temperature and salinity vertical profiles (VOS XBTs and Argo floats), and along-track Sea Level Anomaly observations is performed by a variational scheme (Dobricic and Pinardi, 2008; Storto et al., 2015). Med-PHY extends into the Atlantic Ocean to accurately resolve the dynamical exchange at the Gibraltar Strait, with boundary conditions provided by the CMEMS Global analysis and forecast system products. The upgrade to the increased horizontal resolution at 1/24 degree and the validation of the CMEMS product[1] is thoroughly described in Clementi et al. (2018).

The analysis and forecast product available to CMEMS users for Mediterranean Sea Biogeochemistry[2] consists of 3D daily means of chlorophyll, net primary production, phytoplankton biomass, phosphate, nitrate, oxygen, pH, pCO2. The CMEMS system offers, upon free registration, the access to the 3D fields through the products catalogue and their download via ftp and https protocols (*subsetter* and *directgetfile* download).

## 2.2 Set up of the pre-operational qualification simulation for Med-BIO

The pre-operational qualification run for the Med-BIO component, carried out with MedBFMv2.1, consists of a 2-year re-analysis simulation (1 January 2016 to 31 December 2017), with set up described in the following points.

- The physical ocean (current, temperature, salinity, vertical eddy viscosity) and atmospheric (short wave radiation and wind stress) forcing daily fields are produced by the Med-PHY system and are derived from an equivalent 2-year re-analysis simulation described in Clementi et al. (2018).

- Assimilation of satellite surface chlorophyll concentration derived by the multi-sensor (MODIS and VIIRS) CMEMS product[3] of ocean colour for the Mediterranean Sea is performed by 3DVarBio.

- The initial conditions of biogeochemical variables are set as sub-basin (Fig. 2) climatological profiles computed from in situ data collections (NODC-OGS) described in Lazzari et al. (2016) and Cossarini et al. (2015). A spin-up period of 1 year (2016) repeated for 5 times in perpetual mode is carried out before the start of the simulation.

- The biogeochemical boundary conditions are provided through a Newtonian dumping term which regulates the Atlantic buffer zone western of the Strait of Gibraltar, where the tracer concentrations are relaxed to the seasonally varying profiles. Seasonal profiles of phosphate, nitrate, silicate, dissolved oxygen are derived from an analysis of

---

[1] MEDSEA_ANALYSIS_FORECAST_PHY_006_013
[2] MEDSEA_ANALYSIS_FORECAST_BIO_006_014
[3] OCEANCOLOUR_MED_CHL_L3_NRT_OBSERVATIONS_009_040

climatological MEDAR-MEDATLAS and NODC-OGS datasets, while seasonal profiles of ALK and DIC are obtained from in situ datasets (Huertas et al., 2009; de la Paz et al., 2011; Alvarez et al., 2014).

- Nutrient (nitrogen and phosphorous) loads from 39 rivers (with runoff larger than 50 m³/s) and Dardanelles, which are aligned with the Med-PHY configuration (Fig. 2), are derived from the PERSEUS FP7-287600 project dataset (Deliverable D4.6). The nutrient discharge rates are climatological (averaged over the period 2000-2015) and take into account seasonal variability on a monthly scale.

- Terrestrial inputs of ALK and DIC are estimated on the basis of their typical concentrations per fresh water mass in macro coastal areas of the Mediterranean Sea and the water discharges of the 39 rivers from the PERSEUS dataset. A similar approach holds for the Dardanelles, considered as a river input: the total inflow was derived considering typical water mass concentration of ALK and DIC for Marmara Sea (Copin-Montegut, 1993) multiplied by the net water mass fluxes.

- Atmospheric deposition rates of inorganic nitrogen and phosphorus are set according to the synthesis proposed by Ribera d'Alcalà et al. (2003) and based on measurements of field data (Loye-Pilot et al., 1990; Guerzoni et al., 1999; Herut and Krom, 1996; Cornell et al., 1995; Bergametti et al., 1992). Atmospheric deposition rates of nitrate and phosphate are assumed to be constant in time during the year, but with different values for the western (580 Kt N/yr and 16 Kt P/yr) and eastern (558 Kt N/yr and 21 Kt P/yr) sub-basins. The rates are calculated by averaging the "low" and "high" estimates proposed by Ribera d'Alcalà et al. (2003).

- Atmospheric pCO2 concentration is set equal to the yearly average measured at the Lampedusa station (Artuso et al., 2009) between 1992 and 2017[4], with the 2018 value extrapolated by linear regression.

- Surface evaporation-precipitation effects on dilution and concentration of tracers are directly computed by the OGSTM transport model updated with the non-linear free-surface z*-coordinate configuration.

Further details can be found in the documents available in the CMEMS catalogue (Bolzon et al., 2018; Teruzzi et al., 2019b).

## 2.3 Set up of the operational workflow for Med-BIO

The CMEMS Med-BIO operational workflow runs every Tuesday, starting after the completion of the analysis production cycle of the Med-PHY workflow. The two workflows consist of 7 days of analysis (from T-7 to T-1) one day of hindcast (T0) and 10 days of forecast (from T+1 to T+10, also referred to as T1 to T10) according to the availability of the ECMWF atmospheric forcing. Additionally, in order to maintain enough number of forecast days, Med-BIO performs a new simulation of 10 forecast days on Friday, using the forecast produced by Med-PHY. Boundary conditions in the Atlantic buffer zone, rivers and atmospheric inputs are the same as the pre-operational qualification run, which provided the initial conditions of the operational system at 1 January 2018.

---

[4] http://cdiac.ess-dive.lbl.gov/ftp/trends/co2/lampedus.co2

# 3 Reference datasets for validation

Chlorophyll data are derived by the multi-sensor (MODIS-AQUA and NPP-VIIRS) CMEMS daily product[5] of ocean colour observations for Mediterranean Sea (see Sect. 2.2; Volpe et al., 2007, 2012, 2017) at 1 km spatial resolution. The chlorophyll field combines the estimates of two algorithms for open ocean (case 1) and coastal (case 2) water types. These data are usually released as NRT data within few days from the satellite overpass.

In situ observations of chlorophyll, nitrate and oxygen concentrations are derived by the BGC-Argo floats dataset whose records start from 2013. BGC-Argo floats data are downloaded from the Argo Global Data Assembly Centre webportal and processed following the advanced product quality procedure of Schmechtig and Thierry (2016).

BGC-Argo chlorophyll (Chl) adjusted data are derived from real time (RT) data with a series of corrections: the quenching correction (Xing et al., 2012), a re-calibration at depth (i.e., by imposing zero for Chl values below 600 m), and a tuning correction (i.e., data are further divided by a factor of 2) due to a detection of an error in the manufacturer calibration of Chl fluorometer (Roesler et al., 2017). BGC-Argo nitrate concentrations ($NO_3$) were obtained by using the Johnson and Coletti (2002) algorithm on the raw UV absorption spectrum, then corrected with quality control procedures described in Pasqueron de Fommervault et al. (2015). BGC-Argo oxygen data ($O_2$) are estimated after the application of a quality protocol based on a linear regression constrained to pass through the origin between percent oxygen solubility values derived from the float profiles of $O_2$ and climatological values from the World Ocean Atlas Climatology (Takeshita, 2013; Schmechtig and Thierry, 2016). For the pre-operational period 2016-2017, the total amount of floats and profiles for each variable is given in Tab. 1.

In situ observations of nitrate, phosphate and oxygen derived by the National Oceanographic Data Centre of OGS (NODC-OGS) dataset covering the period 1999-2013 (the list of cruises and datasets is in Lazzari et al., 2016), are used to compute reference climatological profiles for the sub-basins of Fig. 2. In situ observations of DIC, ALK and pH (the list of cruises and dataset sources is in Cossarini et al., 2015) are used to compute reference climatological annual profiles in the sub-basins of Fig. 2. Literature data of net primary production are based on multi-annual simulation (Lazzari et al., 2012), satellite model (Colella, 2006) and in situ estimates (Siokou-Frangou et al., 2010), here used to validate the basin-scale consistency of the corresponding model product.

# 4 Product quality assessment framework

The assessment of the CMEMS Mediterranean Sea biogeochemical model system follows two tasks: pre-operational qualification of the model system and routine (or NRT) validation of forecast products. The aim of the pre-operational assessment is to verify the model consistency, that is its capability of reproducing the salient characteristics of the Mediterranean Sea ecosystem, comparing a short reanalysis run with historical datasets, climatology and literature estimates.

---

[5] OCEANCOLOUR_MED_CHL_L3_NRT_OBSERVATIONS_009_040

The time scale of the comparison ranges from daily to seasonal. On the other hand, the operational assessment relies on the NRT observation availability and aims to evaluate the forecast skills with a temporal scale of days.

## 4.1 Pre-operational quality assessment

The pre-operational qualification is performed at the release of the new CMEMS version using GODAE-like metrics (see Hernandez et al., 2015 for a recent review) applied to the 2-year pre-operational run described in Section 2.2. In particular, the validation consists of "Class 1" metrics, which quantifies the model capability to be consistent with the large-scale climatological description of the ocean processes, and, for a subset of variables (i.e., chlorophyll, nitrate and oxygen), of "Class 4" metrics, which quantify the differences between model and observations at their location and time ("match-ups").

When chlorophyll satellite data are used, the comparison of the model and observations is evaluated before the assimilation (i.e., after 7 days of simulation w.r.t. the previous assimilation cycle) using statistics on the innovation, thus providing a forecast skill metric (Mattern et al., 2018).

For each BGC-Argo float, the vertical profiles of chlorophyll, nitrate and oxygen are matched-up with the model results at the same position and date, producing time series of paired model and observation profiles. Considering the relevance of the seasonal evolution of the chlorophyll vertical profile in the Mediterranean Sea and the importance of analysing the vertical profile as a whole (Lavigne et al., 2015), along with classical observation-model metrics we developed new metrics that synthesize the model capability to reproduce key elements of the vertical profile shape:

- BIAS and root mean square of the difference (RMSD) between model and float of the vertically mixed winter bloom (MWB) depth, defined as the depth at which chlorophyll concentration is 10% of surface concentration during winter (from January to March);
- BIAS and RMSD between model and float of the summer deep chlorophyll maximum (DCM) depth, defined as the depth of the chlorophyll maximum below 40 m during summer (from April to October);
- BIAS and RMSD between model and float of the surface chlorophyll and nitrate concentration (SURF), and of the 0-200 m vertical average of chlorophyll and nitrate (INTG);
- correlation (CORR) between each couple of chlorophyll (oxygen and nitrate) vertical profiles from model and BGC-Argo float;
- BIAS and RMSD between model and float of the depth of the nitracline, defined as the depth (i) where the nitrate concentration is 2 mmol/m$^3$ (NITRCL1), and (ii) corresponding to the maximum nitrate vertical gradient (NITRCL2).

The definitions of DCM and MWB metrics are consistent with the outcomes of Lavigne et al. (2015), who identified some standard shapes for chlorophyll vertical profiles and their temporal distribution from the analysis of a large dataset of fluorescence data in the Mediterranean Sea (see their Figs. 2 and 5). In particular, the summer period defined to estimate the DCM index is based on the consideration that the DCM profile shape is typically observed from April to October. Otherwise,

the choice to limit the estimate of the MWB index from January to March is motivated by the fact that steady depth-decreasing profiles typically occur during that period in different Mediterranean regions. Further, the choice of the 10% criterion for the MWB index was set after a sensitivity analysis varying the threshold between 1 to 10% (not shown), with the 10% value giving results qualitatively consistent with those reported by Lavigne et al. (2015).

The rationale behind the nitracline depth metrics is defining an index useful to track the time evolution of the nitrate profile. Being aware that the choice of a specific value of nitrate concentration may be controversial, we propose two different indexes: the first is based on the depth of the 2 mmol/m$^3$ concentration isopleth (NITRCL1), the second is related to the depth of the maximum nitrate vertical gradient (NITRCL2). According to Manca et al. (2004), the values of nitrate concentration at depth higher than 400 m are around 4-5 mmol/m$^3$ in the eastern basin and 6-7 mmol/m$^3$ in the western, therefore the 2 mmol/m$^3$

isopleth can be considered a consistent threshold to detect the rapid change between the very low concentration typically measured at the surface and the high concentration at depth in all areas of the Mediterranean Sea.

## 4.2 Near-real time (NRT) validation of operational forecast products

The operational skill assessment is performed at weekly frequency considering the results of the previous forecast production cycle and the NRT operational observations (satellite and BGC-Argo floats data) available within one week from the

observation time. Thus, at NRT scale, simulated surface chlorophyll of the first, second and third day of forecast (i.e. forecast lead time of T1, T2 and T3) is compared with the corresponding daily surface chlorophyll from satellite observations, and RMSD and BIAS between model and observations are computed and averaged over the 16 sub-basins. Moreover, all the BGC-Argo float profiles operationally available are compared with the forecast (from T1 to T4) and statistics are reported as weekly time series of RMSD and BIAS between model output of chlorophyll, nitrate and oxygen and observations. The forecast skill

assessment is then compared with the results of the reference pre-operational assessment, which acts as a benchmark.

## 5 Results

## 5.1 Pre-operational qualification run

### 5.1.1 Model consistency

To evaluate the model consistency (GODAE Class 1 metrics) with the general features of the biogeochemistry of the

Mediterranean Sea in terms of chlorophyll, nutrients (nitrate and phosphate), dissolved oxygen, carbonate system variables (DIC, ALK, pCO2, pH), and primary production, model mean fields are compared with different reference datasets.

The MedBFM surface chlorophyll for the period 2016-2017 is compared with satellite data in Fig. 3, while time series of model and satellite data are shown for four selected sub-basins in Fig. 4. The Mediterranean Sea presents a high spatial heterogeneity, with sub-basins characterized by different biogeochemical dynamics (Lazzari et al., 2012). The basin-scale

characteristics, widely described in literature and clearly visible in the maps of Fig. 3 and in the time series of Fig. 4, are the

higher chlorophyll concentrations and the larger seasonal cycle proper of the western sub-basins (e.g. nwm, swm2) with respect to the eastern ones (e.g. ion1, lev2). The MedBFM model correctly simulates the interannual variability observed in the difference between spring blooms in 2016 and in 2017 in swm2, with the former less intense than the latter. A slight model overestimation is observed in alb (Fig. 3), which is probably due to an overestimation of nutrient incoming fluxes at the

Gibraltar Strait. Finally, modelled late winter-early spring surface chlorophyll maxima in nwm appear anticipated of 2-3 weeks w.r.t. satellite ones: this is related to a possible mismatch of the spatial patterns which characterize the temporal succession of deep convection and subsequent stratification and bloom, known to have a very high patchy (i.e., at mesoscale and sub-mesoscale) dynamics in this area (Estrada et al., 2014; Mayot et al., 2017; Severin et al., 2017). The magnitude, timing and spatial pattern of such mesoscale and sub-mesoscale structures might not be completely well resolved, thus resulting in

increased discrepancies with observations.

The MedBFM nitrate and phosphate are in good agreement with the average values and shape of the climatological profiles along the Mediterranean sub-basins (Fig. 5). In particular, the model profiles are within the range of variability of the NODC -OGS climatological profiles (Fig. 5), and the correlation values are generally higher than 0.9 (Fig. 7), which corroborates the very good performance of the MedBFM model in reproducing the deepening of the nutricline and the decreasing concentration

values of the deep layers from the western to the eastern sub-basins. Uncertainty in nwm upper layer nitrate (Fig. 5) is partly related to a possible underestimation of the Ebro/Aude/Rhone rivers input and possibly to the effect of lateral circulation from Alboran Sea and Southern Western Mediterranean surface waters (see Fig. 5, panels "alb" and "swm2").

On average, the RMSD of nitrate is 0.6 mmol/m$^3$ in the upper layers (0-60 m) and around 1 mmol/m$^3$ in the layers below; we observe a general model underestimation of about 30% of the average values at the different depths. Phosphate RMSD is below

0.03 mmol/m$^3$ in the 0-100 m layer, and around 0.04 mmol/m$^3$ in the deeper layers, while BIAS ranges between -0.03 and 0.02 mmol/m$^3$ (Fig. 7). When normalized by the standard deviation of the reference data, the surface layers show the highest uncertainty (i.e., normalised RMSD up to 1 and 1.2 for nitrate and phosphate, respectively) and a relatively low correlation. This is because the surface layers show the lowest concentration values and quite low dispersion of the values among sub-basins. Indeed, simulating nutrient concentration in the layer above the nutricline might be critical, and validation based on

climatological datasets might be not fully appropriate.

Modelled monthly oxygen profiles result pretty well in agreement with the climatological ones and generally within the observed variability (Fig. 5; see also the very high correlation values in Fig. 7), with BIAS and RMSD lower than 11 mmol/m$^3$ in all selected layers (Fig. 7). The RMSDs normalized by the standard deviation range between 0.60 and 1.40 at the different layers, but considering the surface temporal seasonal cycle the normalized RMSD is 0.15. We can observe that the depth of

the relative minimum of oxygen displayed in Fig. 5 is consistent with the cruise data shown by Tanhua et al. (2013): the oxygen minimum layer core in the eastern basin is located below 500 m (sub-basins ion2, ion3 and lev4 in Fig. 5), whilst it is above 500 m in the western basin (see sub-basin nwm in Fig. 5; for the other sub-basins please refer to Teruzzi et al., 2019b).

Figure 6 shows that the model simulates well the vertical structure of DIC and ALK, mostly within the range of variability of the climatological profiles. In particular, it can be noted that the heterogeneity of the vertical profiles of DIC and ALK (i.e.,

the S-shape of western sub-basin profiles, specifically alb and swm2, due to the interaction of surface Atlantic waters and deep Mediterranean waters, and the almost homogeneous vertical profiles for the eastern sub-basins) is fairly well reproduced by the model. For both DIC and ALK, the mean RMSD is around 20 μmol/kg, with higher values for the upper layers. Normalized by the standard deviation of the reference data, the mean errors are 0.40 and 0.70 for ALK and DIC, respectively (Fig. 7).

Correlation values are both higher than 0.7 for almost all layers showing that the basin-wide gradient of carbonate system variables is well captured by the model. The uncertainty of the carbonate system variables strongly reduces at deeper depths and the modelled vertical profiles remain within the climatological variability.

Modelled pH is corroborated using both pH data measured in total scale and reported in situ conditions, and pH data calculated by CO2sys software (Lewis and Wallace, 1998) with available DIC, ALK and other regulatory information (namely:

temperature, salinity and concentration of phosphate and silicate). Modelled pH varies across a 8-8.1 range consistently with the observed eastward and downward positive gradient. The mean error (i.e., averaged RMSD among sub-basins) is around 0.03 in the upper layers and 0.025 in layers below 100 m, which equals almost to the mean variability of data, highlighting that small scale variability of modelled pH cannot be evaluated by the present validation framework.

Finally, modelled pCO2 data can be only qualitatively compared with the reconstructed data using in situ DIC, ALK and the

regulatory information. Along the water column sub-basin profiles, model and reconstructed data show a comparable range of variability. However, it must be noted that the model pCO2 has a large seasonal cycle at surface since the T-dependency of the solubility, while the observations display a lower variability range (Fig. 6) due to the inadequacy of the sampling throughout the seasonal cycle (Cossarini et al., 2015).

Net primary production (NPP) is the measure of the net uptake of carbon by phytoplankton groups (gross primary production

minus fast release processes, e.g., respiration and very labile dissolved organic matter; Vichi et al., 2015). The lack of any extensive dataset of measures of primary production in Mediterranean Sea prevents the application of quantitative metrics for the assessment of the quality of this product. A qualitative assessment of the consistency of the modelled NPP with previous estimates published in scientific literature (Tab. 2) reveals that the simulated relevant gradients between eastern and western regions and averaged NPP values in the different sub-basins are in good agreement with both basin-wide and sub-basin

averages of previous model and satellite assessments. Estimates derived from in situ measurements (Siokou-Frangou et al., 2010) confirm the east to west gradient simulated by the model, though the eastern values appear overestimated by MedBFM.

**5.1.2 Model skill performance**

Skill performance statistics based on a model *vs* observation comparison (GODAE Class 4 metrics) are computed for chlorophyll, nitrate and oxygen, and represent a stricter assessment of the model performance to capture the biogeochemical

temporal dynamics and mesoscale spatial variability.

First, timeseries of RMSD and BIAS of the model-satellite chlorophyll misfit are computed prior the assimilation (i.e. using satellite data that are not yet assimilated), thus representing a short-term (i.e., after 7 days from the previous assimilation cycle) skill forecast metric (Mattern et al. 2018). Then, the mean of the BIAS and RMSD timeseries is calculated for two selected

seasons (i.e., from January to April, WIN, and from June to September, SUM) and reported in Fig. 8, which is completed by the mean spatial standard deviation of observations for each sub-basin.

The western sub-basins have higher uncertainty (i.e. higher RMSD) than the eastern ones, however never exceeding 0.1 mg/m$^3$ on average, and larger during winter period because the variability of the chlorophyll is higher than during summer (Fig. 8). The relatively high values of RMSD in the western sub-basins in winter (nwm in particular) are related to the bloom dynamics, which is estimated 2-3 weeks earlier by the model (Fig. 4). In these areas, blooms are strongly related to the presence of sub-mesoscale local patches, fronts, horizontal circulation structures and local mixing conditions of the water column, as discussed in Section 5.1.1. The large uncertainty in alb, both in winter and summer (Fig. 8), is related to a possible overestimation of the nutrient inflow through Gibraltar Strait. In general, Fig.8 shows that BIAS is positive in winter for the western sub-basins, while is almost negligible in summer for all sub-basins. The value of the chlorophyll RMSD over the Mediterranean Sea, considering the 2016-2017 average, is 0.045 and 0.015 mg/m$^3$ for winter and summer, respectively, while BIAS is 0.015 and -0.005 mg/m$^3$ in winter and in summer. The recently upgraded assimilation scheme that integrates both coastal and open-sea chlorophyll data (Teruzzi et al., 2018) provides a good model performance also in the coastal areas. In these areas the model underestimates the satellite product of about 0.1 mg/m$^3$ in both seasons, and the mean RMSD is about 0.4 mg/m$^3$, with higher values (between 0.5 and 0.9 mg/m$^3$) in areas strongly influenced by coastal processes (not shown). Uncertainty in model prediction in coastal areas is mostly related to the lack of high frequency data for river nutrient discharges which limits the model capability to simulate bloom events triggered by river plume events (Teruzzi et al., 2018).

The comparison of model chlorophyll output with the BGC-Argo floats (Fig. 9 and Tab. 3) provides a skill performance analysis of the model quality in reconstructing the vertical dynamics, integrating the assessment on model surface performances. The Hovmöller diagrams of Fig. 9 show how the time evolution of the model vertical profiles matches up the observations along the corresponding float trajectory. The very good qualitative agreement of the MedBFM model with the BGC-Argo floats is highlighted by the consistent temporal succession of the winter vertically mixed blooms, the onset, the time evolution and the depth of the deep chlorophyll maximum (DCM), which typically establishes during the stratified season. The time series of the new quantitative metrics (defined in Sect. 4.1) computed on the vertical profiles comparison are shown in the lower panels of Fig. 9 for the selected model-float pairs. The agreement between model and float chlorophyll at the surface and its vertical average in the 0-200 m layer is fairly good, with a slight underestimation of the 0-200 m averaged values during winter. Correlation values of the selected float are almost always larger than 0.7, higher during summer and lower in winter. The DCM depth is very well captured by the MedBFM, both in terms of vertical displacement and temporal evolution, and the model MWB depth well performs in 2017, while it appears shallower in 2016.

Averaging the time series of RMSD and BIAS of the new metrics for the aggregated sub-basins (Tab. 3) highlights that the MedBFM model has a very high skill in reproducing the vertical dynamics of the phytoplankton chlorophyll in the 0-200 m layer, considering both the very high spatial heterogeneity of the Mediterranean Sea and the seasonal cycle of the coupled physical-biogeochemical processes. In particular, the correlation between vertical profiles of model and observation ranges from 0.7 to 0.85, with the exception of the Alboran Sea (where only 2 profiles per month are available). The uncertainty of the

DCM position is less than 20 m with a BIAS between -9 and 7 m for the areas with at least 10 float profiles per month, which is a very inspiring result considering that the model vertical discretization is about 6-8 m for the layers around the depth of 80-120 m. The depth of the MWB is not computable and reliable for some of the sub-basins. However, for those having more than 10 float profiles per month, it has an absolute BIAS ranging from 30 to 40 m and a RMSD ranging from 40 to 50 m.

Considering the constraint in the definition of the MWB depth and the vertical discretization of the model, the application of such index to floats data may indeed originate some inconsistency (as shown for winter 2016 in Fig. 9), and under- or overestimations and uncertainty of a few decametres (see Tab. 3). Despite these limitations, we consider the MWB as a feasible and informative metric alongside the DCM metrics to characterize the seasonal chlorophyll profile evolution.

The averaged vertical values show that the model generally underestimates the content of chlorophyll with respect the BGC-

Argo floats measurements, which appears in contrast with the general assessment of model overestimation for the winter period w.r.t. the satellite data. Triple collocation method, as proposed by Mignot et al. (2019), might be applied to investigate possible off-sets and random errors among multi-platform datasets at regional/local scale. Nevertheless, the RMSD of the 0-200 m vertical averages remains lower than 0.1 mg/m$^3$ in all the aggregated sub-basins.

The comparison of model nitrate with the BGC-Argo float measurements allows to evaluate the skill of the MedBFM to

simulate key coupled physical-biogeochemical processes (i.e., water column nutrient content, nitracline, and effect of winter mixing and summer stratification on the shape of nitrate profile; metrics defined in Sect. 4.1). Qualitatively, we observe a general good model performance in simulating the shape of the profile (i.e. correlation values), the temporal evolution of the 0-200 m averaged values and of the nitracline depth of the selected float (Fig. 10). The model NITRCL1 and 2 perform generally good, however, it can be observed that in the period April-July 2017 the NITRCL2 appears much shallower than

what estimated by the float data. The two indexes show different aspects of the nitrate profile evolution, justifying their use to provide indications aimed to monitor the model error behaviour.

Tab. 4 shows the nitrate metrics of the 8 floats, averaged over the aggregated sub-basins: even if the scarcity of the profiles possibly limits the generalization of the results, our validation framework highlights that the MedBFM model system shows excellent performance in simulating the shape of profiles and the seasonal evolution of the mesoscale dynamics affecting the

nitrate field. In particular, Tab. 4 reports that the mean value of nitrate on the 0-200 m layer is very well simulated, with BIAS ranging from 0.04 to -0.68 mmol/m$^3$ and RMSD generally smaller than 1 mmol/m$^3$; the correlation is always higher than 0.9 and the depth of the nitracline is simulated with an uncertainty lower than 40 m. Further, accordingly with BGC-Argo floats observations (Tab. 4), the MedBFM reproduces fairly well the Mediterranean basin scale heterogeneity with a nitracline at around 60-100 m in the western sub-basins and below 110 m in the eastern sub-basins.

The qualitative comparison of modelled oxygen with a selected BGC-Argo float (Fig. 11) shows the MedBFM skill to simulate the sequence of physical-biogeochemical processes of the oxygen dynamics, such as the effect of ventilation during winter, the production of an oxygen maximum at the layer of the DCM due to the intense phytoplankton production during spring and summer, and the minimum of oxygen concentration at surface during summer and autumn due to decrease of solubility and presence of consumption terms (defined as respiration terms by bacteria and plankton community: 4 phytoplankton and 4

zooplankton groups). Interestingly, the depth of ventilation has a clear interannual variability, as shown by the higher values of oxygen below the 100 m depth in the event of December 2016 – January 2017 with respect to the previous year. The quantitative comparison between all the available floats data and model results is summarized by the statistics of Tab. 5, showing a general model overestimation of about 15 mmol/m$^3$ at surface, increasing with depth to about 20-25 mmol/m$^3$. The

Adriatic Sea (data from 1 float only) shows a much lower discrepancy of around 5-10 mmol/m$^3$.

Discrepancies at surface might be due to solubility calculation, whereas at depth to inaccuracies of the initial conditions or to excess of production. However, considering the modelled bias error in temperature and salinity at surface of -0.23°C and 0.01, respectively (Clementi et al., 2018), and under the hypothesis of oxygen solubility at surface, the BIAS for the modelled oxygen (i.e., calculated using the formulations of Weiss, 1970, and of Garcia and Gordon, 1992) should not exceed 1-1.5

mmol/m$^3$ throughout the year. On the other hand, the on-going improvement of quality control procedures (Johnson et al., 2017) and the need for reprocessing might have an impact on the accuracy of archived oxygen data. Only very recently a new product quality control procedure (following Bittig et al., 2018 and Thierry et al., 2018) has started to be implemented for oxygen data to correct biases on sensors: to the best of our knowledge, it is not yet available for all floats in the Mediterranean Sea. Thus, this comparison must be considered cautionary; nevertheless it provides a qualitative indication of the model

behaviour to capture spatial and temporal oxygen dynamics.

### 5.2 Near-real time forecast skill performance

The near-real time (NRT) skill performance of the operational forecast system aims at delivering sustained on-line information on the quality of Med-BIO biogeochemical forecast products, i.e., firstly identifying main biases and possible suspicious trends in the forecasts, and secondly establishing that the accuracy remains within the assessed ranges. The NRT validation activities

are performed using GODAE Class 4 metrics with available satellite data for the first three days of forecast (T1-T3 lead time) and BGC-Argo floats observations for the first four days of forecast (T1-T4 lead time) using the same metrics computed for the pre-operational run, that provides the benchmarks of the accuracy level. Online resources for such metrics (RMSD and BIAS between model and observations averaged over the sub-basins) are updated quarterly on the official CMEMS validation webpage[6] and weekly on the regional Mediterranean validation website managed by OGS[7].

Figure 12 reports the RMSD between NRT daily L3 multi-sensor satellite data (see details in Sect. 2.2) and the first three days of forecast for selected sub-basins since April 2018 (i.e., the start of the last version of the CMEMS Med-BIO system at the time of writing). Similarly to the pre-qualification run (Fig. 8), the forecast skill metrics are characterized by a seasonal and spatial variability that basically reflects the chlorophyll spatial and temporal variability. For the period reported in Fig. 12, the performance of the first day of forecast is generally better than the benchmark references, while it decreases for the second

and third day of forecast. Indeed, the average of the RMSD over the 16 sub-basins is 0.018, 0.034 and 0.041 mg/m$^3$ for the

---

[6] Available at http://marine.copernicus.eu/services-portfolio/scientific-quality/
[7] Available at http://medeaf.inogs.it/nrt-validation/

first, second and third day of forecast, respectively. The high variability of the RMSD statistics from one day to another is basically related to the daily varying number of available pixels, due to the cloud cover and its spatial distribution.

To provide a monitoring of the quality of the NRT forecast with respect to a seasonal reference defined by the pre-operational qualification run, Figure 13 shows the distribution of the available BGC-Argo data matched up with the forecast data of

chlorophyll, nitrate and oxygen basin-averaged on different vertical layers for the first four days of forecast, and a season-based benchmark represented by the results from the pre-operational run. In general, the forecast data are within the variability of the seasonal benchmark (in this case, the period from May to August). Indeed, the overall RMSD metrics of the forecast skill of chlorophyll and nitrate are always lower than the values estimated for the pre-operational run (Tab. 6), while the RMSD statistics of oxygen forecast highlight the bias in the lower layers and are slightly higher than the computed RMSD for the pre-

operational run. We can observe that floats oxygen concentrations in the subsurface layer (100-150 m) are lower than 180 mmol/m$^3$, which appears quite anomalous for the Mediterranean Sea (see Manca et al., 2004, and also Tanhua et al., 2013), thus conveying a suspect bias of the oxygen data retrieved from BGC-Argo repository, as already discussed in Section 5.1.2 for Fig. 11.

The RMSDs of the four forecast days (Tab. 6) remain within a range of ±25%, generally showing that the quality of

biogeochemical forecast does not significantly degrade during the first week. More precisely, chlorophyll and oxygen RMSD of T3 and T4 are slightly larger than T1, while nitrate RMSD of the last forecast days is lower than T1. However, considering the very low number of available data (few tens in the 5 months considered) and the fact that BGC-Argo floats data may exhibit wide oscillations over subsequent profiles (as shown in Fig. 10), the differences of skill performance statistics from one day of forecast to another might be considered cautionary.

**6 Discussion**

This work presents the last achievements in the operational biogeochemical component for the Mediterranean Sea delivered by CMEMS. The MedBFM model system has been integrated with the last scientific achievements of the BFM model (Cossarini et al., 2015; Lazzari et al., 2016), the 3DVarBio assimilation scheme (Teruzzi et al., 2018, 2019a) and the non-linear free surface and volume vertical layer parameterization of the transport operator of the OGSTM model (Salon et al.,

25    2018).

The Med-BIO system has followed the developments of the EU operational marine services (Le Traon et al., 2017), starting from its first version (Lazzari et al., 2010) deployed within the MERSEA project (2004-2008; GMES implementation phase), becoming pre-operational during MyOcean projects series (2009-2015; GMES demonstration and pre-operational phase) and finally establishing a regular and validated operational product delivery in CMEMS (GMES operational phase). Across this

10-year period, the quality of the Med-BIO products has significantly increased (Fig. 14, quality assessed by the RMSD of the surface chlorophyll concentration, the only product variable that has been consistently validated since the beginning of the

Med-BIO activity), with a continuous improvement which took advantage from the implementation of the data assimilation, the increased horizontal resolution, and the evolutions in the physical component of the Med-MFC system.

The Med-BIO off-line coupling with Med-PHY was outlined since the preliminary work of Lazzari et al. (2010) and has allowed for distinctive developments of the different components. Further, the alignment between physical and biogeochemical
models in terms of same horizontal resolution, bathymetry, boundaries (number and position of rivers) and surface forcing (e.g., z* parameterization), a requisite of the CMEMS framework, guarantees the consistency of the results (as shown by the recent improvement of the performance after April 2018, Fig. 14). Other studies demonstrated that off-line coupling does not affect the transport of biogeochemical tracers when the sub-mesoscale physics is degraded to mesoscale (Levy et al., 2012). Further improvements of the Med-BIO biogeochemical model system in terms of physical-biogeochemical consistency at local
scale are expected with the foreseen implementation of the assimilation of the BGC-Argo floats data (Cossarini et al., 2019), which has shown the improvement of the model solution due to the increased consistency of vertical dynamics by the assimilation of the physical and biogeochemical profiles at the same time and position. This result highlights the importance of the joint physical and biogeochemical assimilation, that has been recently demonstrated in a twin experiment to provide superior results with respect to any uncoupled assimilation configuration (Yu et al., 2018).

Communicating the uncertainty is a critical point: it helps the users to properly interpret the validity of the forecast products, even when the forecast actually fails, and to minimize any problem created by the misuse (and misinterpretation) of them (Stow et al., 2009; Payne et al., 2017). The communication of the level of uncertainty of the biogeochemical products remains an open issue for the scarcity of reference NRT biogeochemical observations available and for the complexity of
biogeochemical models, which may have tens of variables but only a few can be validated. Further, regional operational models have reached the limit of the sub-mesoscale, which is not adequately sampled by observational systems (Hernandez et al., 2018). As an example, the number of dissolved oxygen observations used to build Fig. 6 is almost one fifth of those available for phosphate (Teruzzi et al., 2019b), therefore the reliability of validation using the climatological profiles might be lower and even less for the surface values, since dissolved oxygen exhibits a significant seasonal and high frequency cycles due to
the air-sea exchanges mediated by solubility.

We show that depending on the variables, different uncertainty levels can be provided on the basis of the availability of reference data. In this context, the validation analysis provides a "degree of confirmation" (Oreskes et al., 1994) with respect to the different scales of variability derived from the available observations. GODAE Class 1 metrics show that the model is consistent (in terms of chlorophyll, nitrate, phosphate, oxygen, dissolved inorganic carbon, alkalinity and pH) in reproducing
the vertical profile climatology at sub-basin scale (Figs. 3 to 8). The comparison of model primary production with available basin-wide estimates and literature collection can only provide a consistency confirmation of the model estimates at the basin and annual scales. Then, we also demonstrate that GODAE Class 4 metrics are feasible and provide more rigorous skill performance down to the scale of week and mesoscale, but only for a limited number of variables (see Figs. 9 to 13). Regarding data availability, satellite chlorophyll estimates represent the most reliable source of NRT data, which, however, allows to

investigate only the cloud-free surface of the ocean. The novel BGC-Argo floats dataset empowered us to design new skill metrics, showing the capability of the MedBFM model to reproduce the temporal evolution of the vertical dynamics of the phytoplankton, nitrate and oxygen, and to assess key ecosystem processes in the Mediterranean Sea. The novel metrics based on BGC-Argo data disclose new and important perspectives for the model validation in the Mediterranean Sea, also considering

its very high spatial heterogeneity and the seasonal variability of the coupled physical-biogeochemical processes. However, some cautions should be taken before generalizing the conclusions, since the relatively poor BGC-Argo floats coverage in some areas and the on-going improvement of product quality procedures of the BGC-Argo data (Johnson et al., 2017). Moreover, the relevance of the representation error (Hernandez et al., 2018) becomes stricter with BGC-Argo data, since the skill performance analysis is based on comparing a model output with grid cells 3-4 km wide and O(10) meter thick with point-

like profiles of few meters of resolution, thus the model may miss part of the spatial-temporal scales present in the observations (Oke and Sakov, 2008).

Considering the NRT evaluation, Figures 12 and 13 show that the uncertainty of the forecast products is of the same order (and in some occasions even lower) than the pre-operational run, and that the performance decreases slightly from the first day of forecast to the following ones. This is related to both the uncertainty due to the intrinsic error of the biogeochemical model

and the decrease of the performance of the physical variable forecasts driving the biogeochemical ones (see as reference, Clementi at al., 2018, and the CMEMS quarterly validation statistics[8]). Furthermore, comparing NRT metrics with a seasonal benchmark can highlight anomalous model behaviors that may constitute an operational monitoring system to alert the users and the researcher staff that model performance is worsening or a specific event is occurring, thus conveying useful information to investigate possible causes.

We show that the error statistics, in terms of RMSD, are proportional to the variability of the variables. It is shown for surface chlorophyll (Fig. 8) and it can be also noted from BGC-Argo derived statistics: sub-basins characterized by higher variability have higher error (Tab. 3). As a result, the performance analysis shows that the western regions have, in general, largest variability and lower performance, specifically during the winter season. Thus, to rationalize the costs of observing systems

(Cristini et al., 2016), it may be more efficient to sustain the observing systems with high-frequency observations in high-variability areas. On the other hand, given that fields variability may be related to local physical and biogeochemical processes (e.g. vertical mixing, coastal effects due to strong topographic gradients or terrestrial inputs), the reduction of the model representativeness error can benefit from a more collaborative evolution of the coupled physical-biogeochemical systems, both in terms of process modeling or coupled data assimilation (Fennel et al., 2019).

The present validation framework uses an *a priori* subdivision which considers the biogeographic approach of D'Ortenzio and Ribera d'Alcalà (2009), and the subsequent refinement proposed by Lazzari et al. (2012) which showed different characterizations according to the Longhurst paradigm. The recent review of Ayata et al. (2018) discusses the variations of the

---

[8] Available at http://marine.copernicus.eu/services-portfolio/scientific-quality/.

Mediterranean Sea subdivision found in literature, highlighting regions with relatively homogeneous conditions and some heterogeneous regions featuring significant mesoscale activity. Our validation approach demonstrates the importance to provide model uncertainty estimation at different spatial and temporal scales, emphasizing the model capability to reproduce specific processes and their intensity in different areas, while computing metrics over the sub-basins allows to synthetize the heterogeneity of the Mediterranean Sea, justifying *a posteriori* our sensible definition of the 16 sub-basins. In fact, the comparison of nutrients profiles of Fig. 5 highlights the satisfactory model performance in reproducing the mean spatial gradients and the possible anomalies, such as the underestimation of upper layer nitrate in nwm sub-basin. Moreover, the use of simple indexes, such as means and standard deviation, and of functional spatio-temporal subdivisions increase the readability of the uncertainty communication, which responds also to the request for a user oriented evolution of the validation framework in operational systems (Hernandez et al., 2018).

Our validation results point out a number of strengths and weaknesses of the CMEMS Mediterranean forecasting biogeochemical system. A strength is that the MedBFM is operationally in place and provides validated and reliable ecosystem products consistently with the physical ones (Clementi et al., 2018).

The system can also provide important feedbacks to the observing autonomous systems. Indeed, the NRT comparison of BGC-Argo floats data with the forecast outputs w.r.t to the seasonal benchmarking might be beneficial for an additional QC procedure for detecting anomalous observations that the present QC fails to detect, as proposed for physical variables measured by Argo systems (Ingleby and Huddleston, 2007). As an example, the Class 4 metrics applied to BGC-Argo oxygen (Fig. 13, third column) shows a systematic bias which does not appear when contrasted with the Class 1 validation (Figs. 5 and 7), thus pointing out the opportunity of a possible revision of some model formulations or product quality procedure of BGC-Argo oxygen data in the Mediterranean Sea. A specific investigation focused on the oxygen validation framework and the analysis of the oxygen variability simulated by the MedBFM model is in preparation.

Another positive aspect of our work is that, to the best of our knowledge, this is one of the first times that a consistent validation procedure provides sustainable guidelines following GODAE metrics for operational marine biogeochemistry exploiting the BGC-Argo floats data (Hernandez et al., 2018). Our novel metrics (Figs. 9 and 10) provided indications of the model skill performance on some key biogeochemical processes (DCM, nutricline depth), thus setting an advancement to what described in Hernandez et al. (2015) for NRT assessment of biogeochemical operational forecast and maximizing the values of the available NRT biogeochemical observations (She et al., 2016). In perspective, the integration of BGC-Argo within operational ocean forecasting systems in terms of data assimilation (see Cossarini et al., 2019) becomes strategical for an in-depth study of the interior of the sea and its dynamics. Moreover, considering that BGC-Argo floats also provide profiles of physical quantities (i.e., radiometric quantities – PAR – and temperature), an analysis of specific physical (e.g., MLD, euphotic zone depth) and biogeochemical (e.g., NITRCL, MWB, DCM) indexes that can reveal relationships between the shape and/or intensity of the profiles and the underlying dynamics would allow to further delve into coupled vertical physical-biogeochemical processes. In such a view, our work provides a first step to identify and quantify several functional

biogeochemical indexes. Nonetheless, a critical point remains the availability of a sufficient amount of profiles for variables like nitrate and oxygen, which may allow for statistically significant analysis.

Concerning the weakness of MedBFM, we may point out the reduction in performance close to the domain boundaries at Gibraltar and Dardanelles Straits and in the coastal areas. The observed overestimation of chlorophyll, and thus of productivity and phytoplankton biomass, in Alboran Sea (see Figs. 3 and 8) can be related to an incorrect parameterization of the biogeochemical fluxes through Gibraltar Strait or to the effect of vertical mixing. Inconsistent physical-biogeochemical data assimilation might generate incompatible density and nutrients profiles that may generate an extra amount of vertical flux of nutrient in this highly dynamical area, thus enhancing its productivity. Increase of nutrient availability along isocline surfaces has been observed by Raghukumar et al. (2015) suggesting this as a possible cause of increase of productivity in oligotrophic areas. The upgrade of MedBFM boundary conditions (at the Gibraltar Strait, Med-PHY is coupled with the CMEMS global product while Med-BIO uses climatological biogeochemical value) with high frequency values, and the extension of the Atlantic buffer zone, could improve the model performance in this area.

For the coastal areas, the increased resolution to 1/24° cannot fully balance the use of low frequency data of biogeochemical terrestrial inputs (i.e. nutrients and carbonate system estimates from climatological databases). Thus, some quality decrease is observed even though data assimilation of coastal chlorophyll from satellite can partly reduce this deficiency (Teruzzi et al., 2018). Operational or at least higher frequency coastal data for rivers and the inclusion of Dardanelles as an open boundary condition are requested to account for the user needs of reliable products in coastal areas.

## 7 Conclusions

The present work evaluates the skill performance of the CMEMS Mediterranean Biogeochemistry component (Med-BIO) determining the quality of the CMEMS biogeochemical products on the basis of two complementary phases: 1) the pre-operational qualification run (2016-2017), and 2) the operational workflow (started in April 2018 for MedBFMv2.1).

Using different observation reference datasets (from satellite, literature, climatology, BGC-Argo floats), GODAE Class 1 and 4 metrics have been applied to the MedBFM model system in order to quantify its consistency in simulating the key features of the Mediterranean biogeochemistry, and its accuracy to routinely reproduce the observations at their specific time and locations. New metrics specifically designed to exploit the richness of BGC-Argo floats database and to evaluate the model capability to reproduce the key elements of the vertical profiles of chlorophyll and nitrate have been proposed. Main results can be here summarized:

- MedBFM is consistent in reproducing the general characteristics of biogeochemistry in Mediterranean Sea, and the CMEMS Med-BIO products are well within the climatological variability; quantified correlation values are higher than 0.9 and 0.7 for nutrients and carbonate system products, respectively.
- The level of accuracy of the different Med-BIO products depends on the kind of variable, the availability of reference data, the sub-basin and the season.

- Novel Class 4 metrics based on the model match-up with BGC-Argo floats data represent a useful tool to quantify the capability of a biogeochemical model to reproduce key elements of the biogeochemical processes along the water column (depth of deep chlorophyll maximum, mixed winter bloom, nutricline). For MedBFM, correlation is generally higher than 0.7/0.9 for vertical profiles of chlorophyll/nitrate, and errors (as RMSD) in reproducing the key depths ranges between 12 and 50 m.

- NRT validation of Med-BIO forecast products have been performed for chlorophyll, nitrate and oxygen from April 2018, showing a slight decrease of forecast skill performance after 1 or 2 days for surface chlorophyll and a not unique identified pattern when BGC-Argo data are used. Nevertheless, the forecast skill performance remains at the same level as the benchmark within the first week of forecast.

Even if the use of BGC-Argo floats significantly discloses new perspectives for operational biogeochemical model validation, some cautions should be considered before generalizing the conclusions, due to the relatively poor BGC-Argo coverage in some areas of the Mediterranean Sea and the on-going improvement of product quality procedures of the BGC-Argo data. Robust statistics require much longer time series of data and a larger number of BGC-Argo floats, which is becoming an urgent request for the observing systems to be used in operational biogeochemical oceanography (for both validation and assimilation purposes).

Finally, the validation metrics here presented provides indications of some weaknesses of the Med-BIO (e.g. limited dynamics in coastal areas, Gibraltar boundary and sub-mesoscale effects on phytoplankton dynamics in western area) that will lead to future developments. Nevertheless, the validation results support not only the accuracy of the CMEMS Med-BIO products, but also the consistency of the MedBFM model system to simulate the fundamental coupled physical-biogeochemical processes, which is corroborated at the mesoscale and weekly scale.

**Author contribution**

SS and GC conceived the ideas of the work, the formulation of main research goals, the investigation, and the methodology. SS was in charge of the supervision of the manuscript preparation, preparing the original draft with contributions from all co-authors. GC and SS conducted the formal analysis supported by LF. GB was in charge of the data curation, LF prepared most of the figures. GC, PL, AT and LF contributed to review the original draft.

GC is responsible of the CMEMS Med-BIO system; GB is in charge of the operational workflow; LF conducts the NRT operational validation; GC, GB, AT, SS and PL worked on the upgrade of the CMEMS Med-BIO system. CS, AC, GC and SS contributed to the funding acquisition.

**Acknowledgements**

This study has been conducted using E.U. Copernicus Marine Service Information. The authors thank Dr. V. Di Biagio (OGS) for fruitful discussion concerning the review of biogeochemical model applications in Mediterranean Sea, and Dr. A. Marani (CINECA) for the technical support provided in the MedBFM production workflow. The authors want also to thank the two anonymous reviewers for their very useful comments.

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

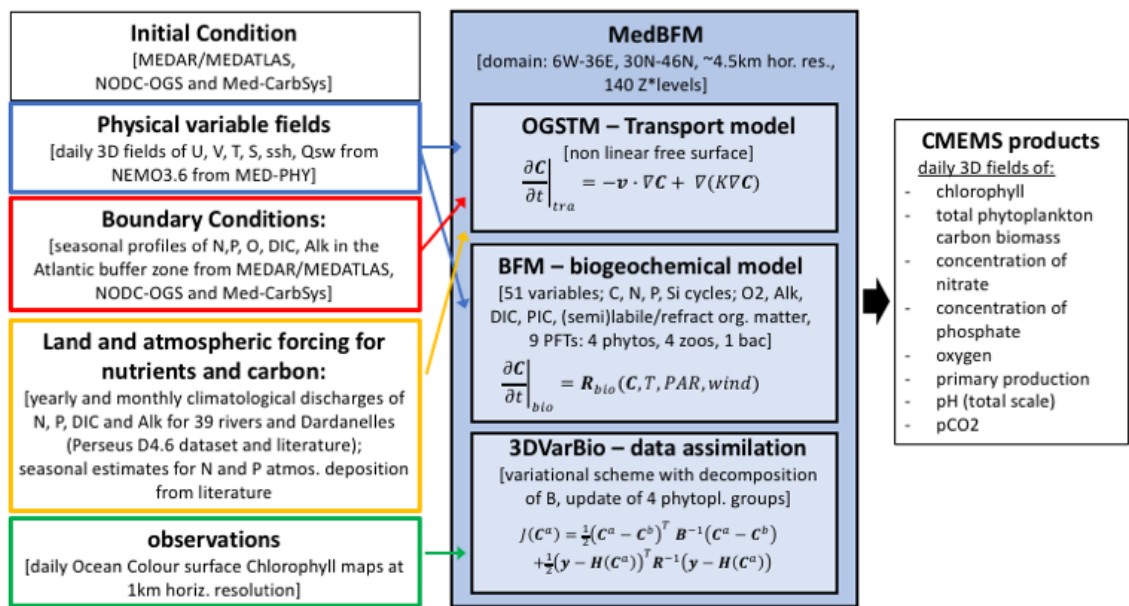

**Figure 1: The MedBFMv2.1 model system and interfaces with other components of CMEMS and external forcing data.**

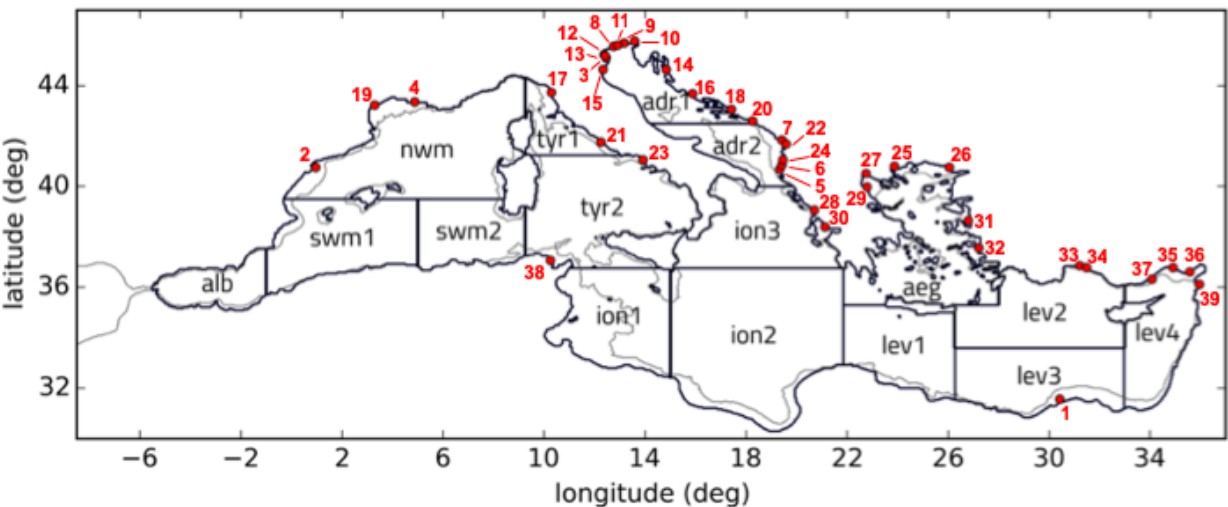

Figure 2: Subdivision of the model domain in sub-basins used for the validation of the qualification run. According to data availability and to ensure consistency and robustness of the metrics, different subsets of the sub-basins or some combinations among them can be used for the different metrics: lev = lev1+lev2+lev3+lev4; ion = ion1+ion2+ion3; tyr = tyr1+tyr2; adr = adr1+adr2; swm = swm1+swm2. The grey line defines the bathymetric contour at 200 m. Red dots with numbers correspond to river mouths positions: Nile (1), Ebro (2), Po (3), Rhone (4), Vjosë (5), Seman (6), Buna/Bojana (7), Piave (8), Tagliamento (9), Soca/Isonzo (10), Livenza (11), Brenta-Bacchiglione (12), Adige (13), Lika (14), Reno (15), Krka (16), Arno (17), Nerveta (18), Aude (19), Trebisjnica (20), Tevere (21), Mati (22), Volturno (23), Shkumbini (24), Struma/Strymonas (25), Meric/Evros/Maritsa (26), Axios/Vadar (27), Arachtos (28), Pinios (29), Acheloos (30), Gediz (31), Buyuk Menderes (32), Kopru (33), Manavgat (34), Seyhan (35), Ceyhan (36), Gosku (37), Medjerda (38), Asi/Orontes (39).

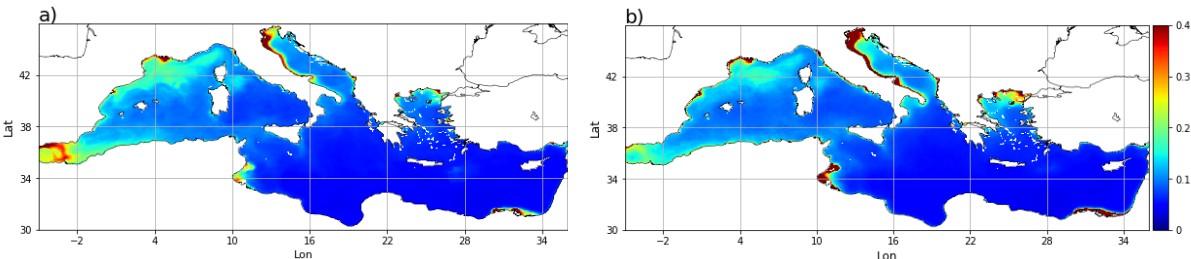

**Figure 3: Averaged annual maps of surface chlorophyll (mg/m³) from 2016-2017 qualification run (a) and from NRT multi-sensor satellite (b).**

**Figure 4: Model (black line, with standard deviation in black dots) and satellite (green dots, with standard deviation covering the light green area) time series of mean surface chlorophyll concentration in open sea areas in four selected sub-basins of Fig. 2.**

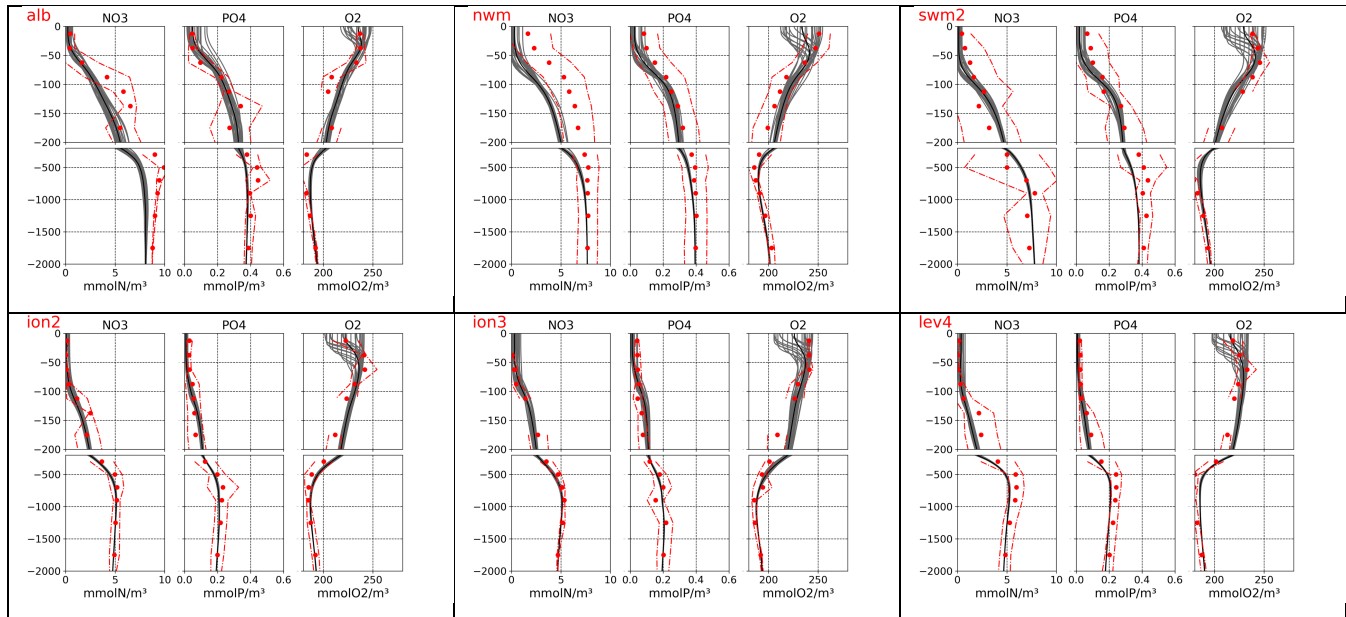

**Figure 5: Monthly (grey lines) and mean (black lines) vertical profiles from the qualification run for selected sub-basins of Fig. 2 compared with climatological profiles (red dots) and variability ranges (one standard deviation, red lines) of nitrate, phosphate and dissolved oxygen retrieved from NODC-OGS dataset.**

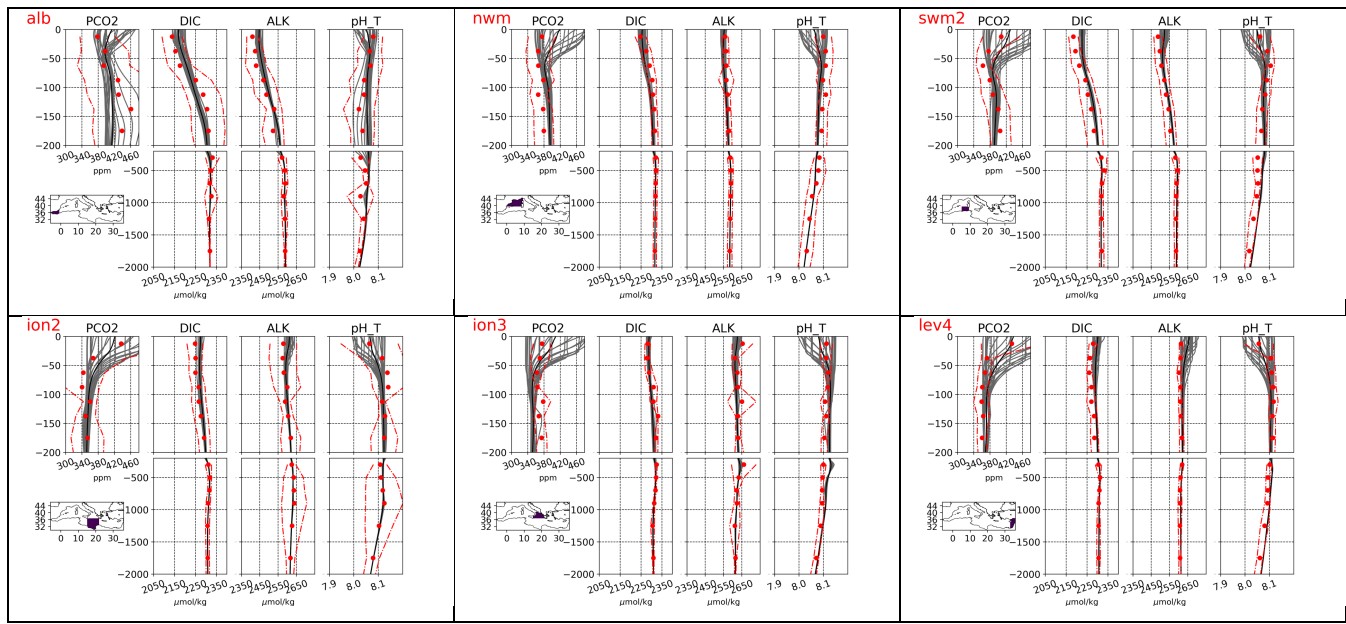

5    **Figure 6: Monthly (grey lines) and mean (black lines) vertical profiles from the qualification run for selected sub-basins of Fig. 2 compared with climatological profiles (red dots) and variability ranges (one standard deviation, red lines) of Dissolved Inorganic Carbon (DIC), alkalinity (ALK), pH in total scale and in situ condition (pH_T) and carbon dioxide partial pressure (pCO2). Climatological data for pH_T and pCO2 are reconstructed using CO2Sys software (Lewis and Wallace, 1998).**

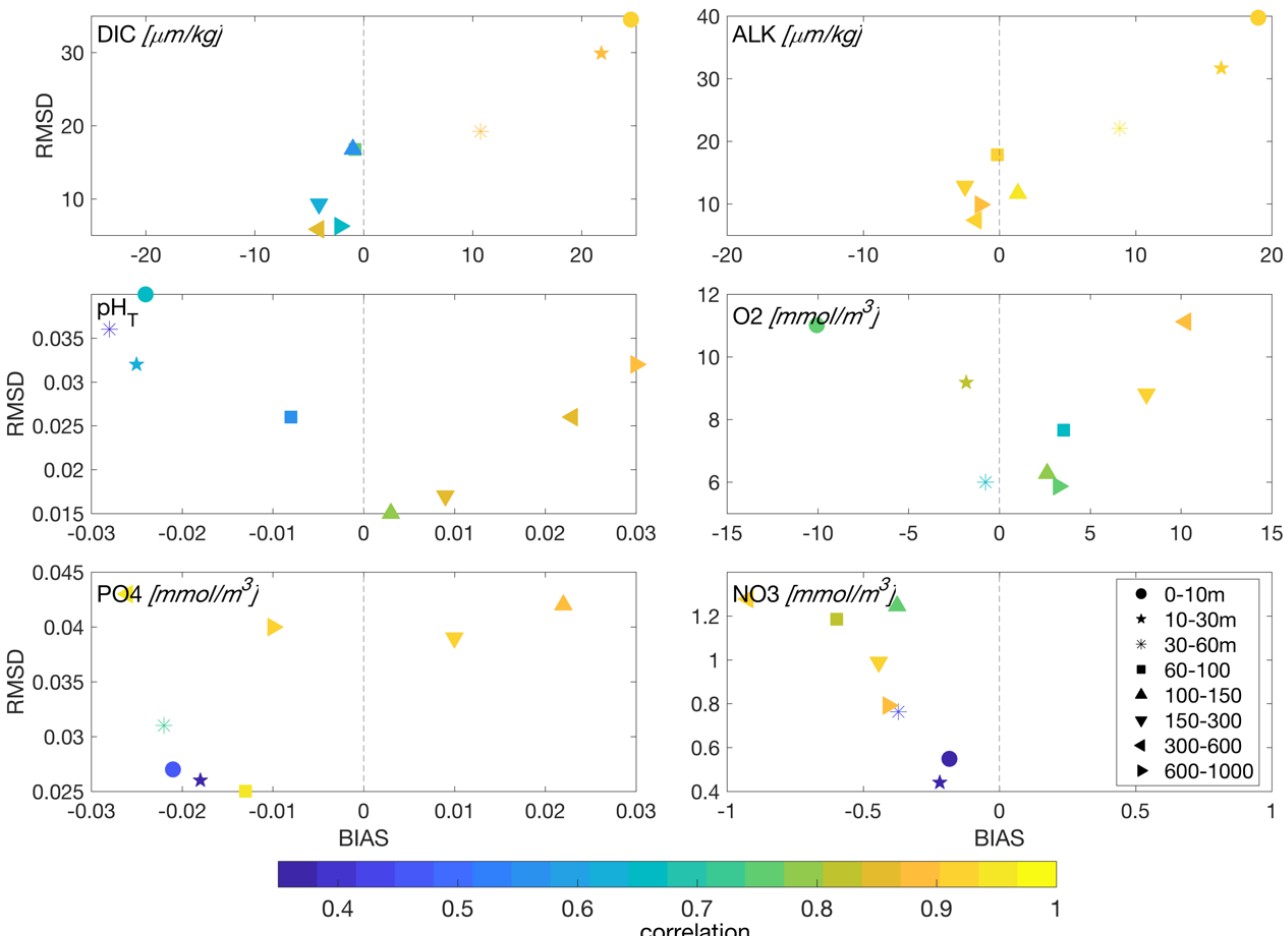

**Figure 7: Target diagrams and correlation (values in colour shading) between model and climatology for the different layers (symbols) and for the selected variables: alkalinity (ALK), dissolved inorganic carbon (DIC), oxygen (OXY), phosphate (PO4), nitrate (NO3) and pH reported in total scale (pH_T).**

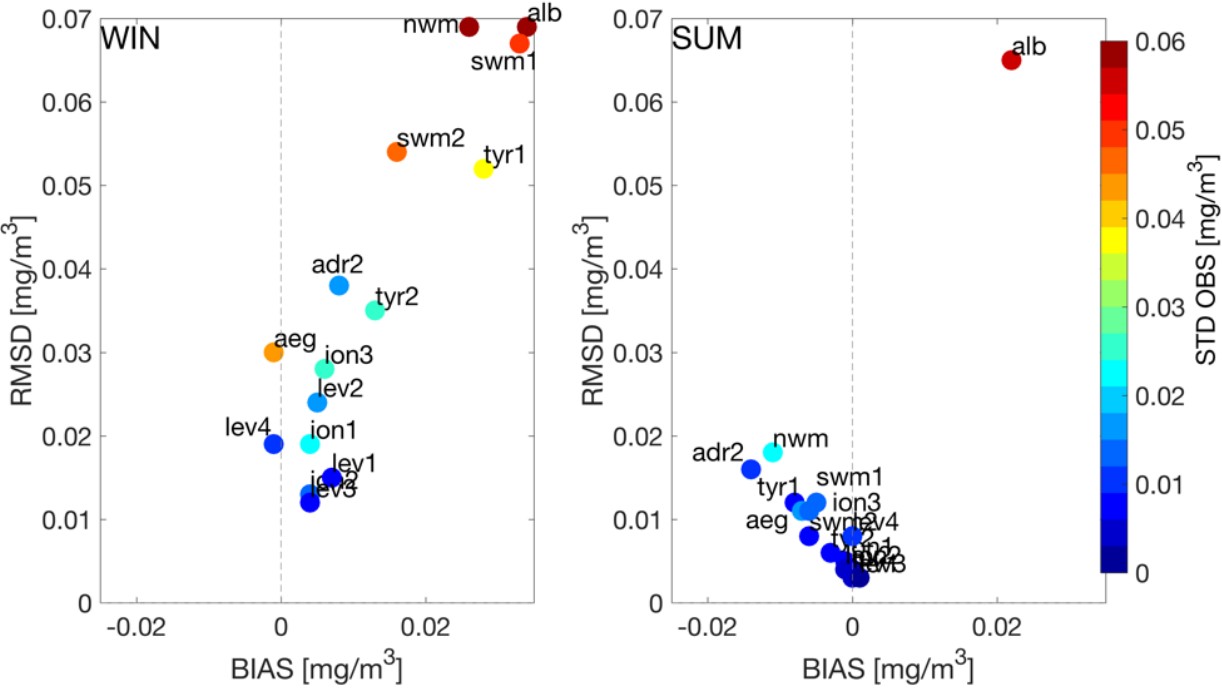

**Figure 8: Target diagrams of the model and satellite chlorophyll comparison and standard deviation of observations (in colour shading) for two periods: January to April (WIN, left) and June to September (SUM, right). For sake of readability, an offset in the values in WIN of [RMSD, BIAS] for alb and of [RMSD] for nwm (respectively equal to [0.17, 0.09] mg/m$^3$, and [0.1] mg/m$^3$) has been applied to include the dots within the plot.**

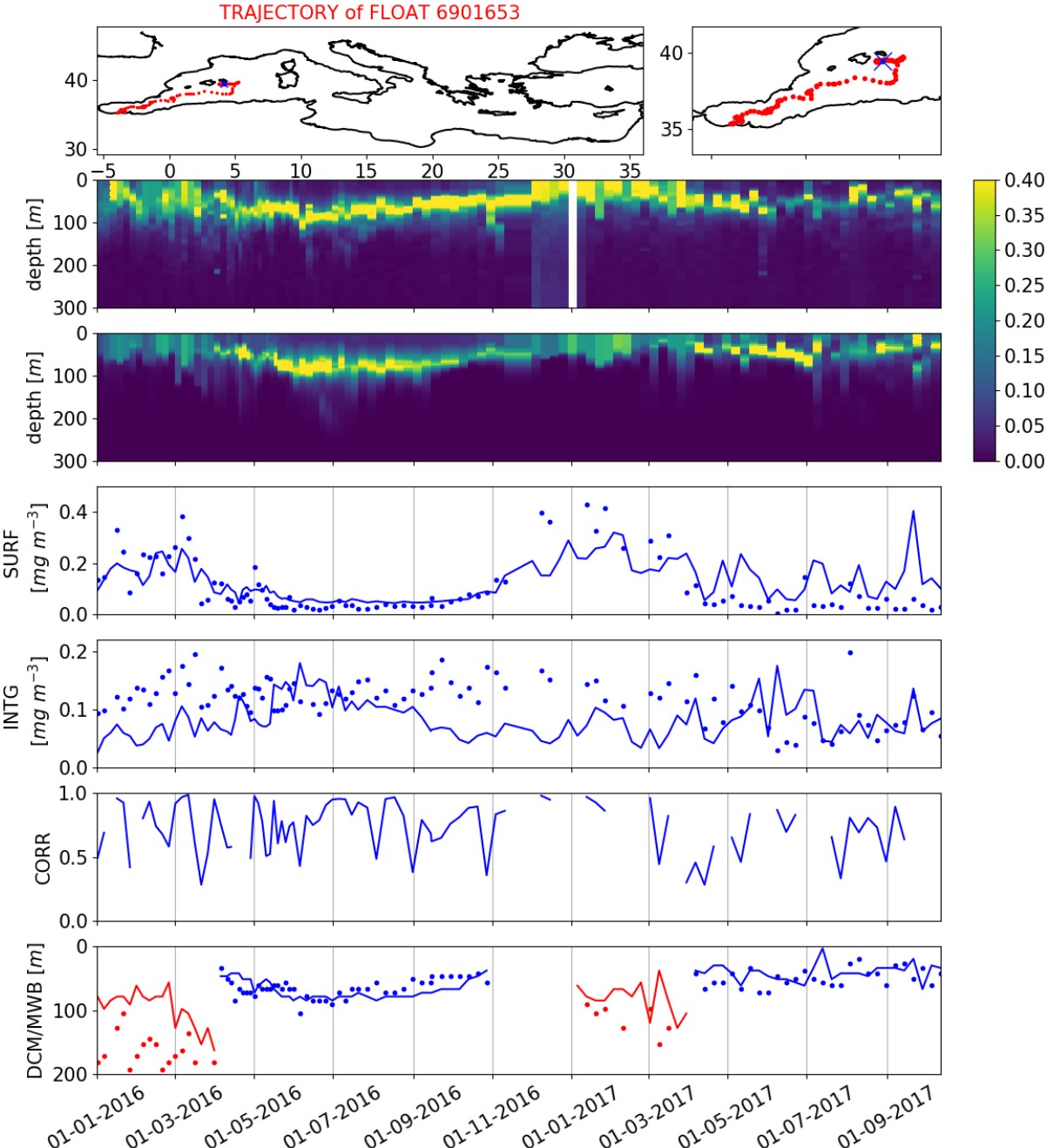

**Figure 9: Time evolution of BGC-Argo float 6901653. Top panel: trajectory of the BGC-Argo float (red dots), with deployment position (blue cross); Hovmöller diagrams of chlorophyll concentration (mg/m³) from float data (2ⁿᵈ panel) and model outputs (3ʳᵈ panel) matched-up with float position for the period 2016-2017. Computation of selected skill indexes for model (solid line) and float data (dots): surface chlorophyll (SURF, 4ᵗʰ panel) and 0-200 m vertically averaged chlorophyll (INTG, 5ᵗʰ panel), correlation between vertical profiles (CORR, 6ᵗʰ panel), depth of the deep chlorophyll maximum (DCM, blue) and depth of the mixed layer bloom in winter (MWB, red; bottom panel).**

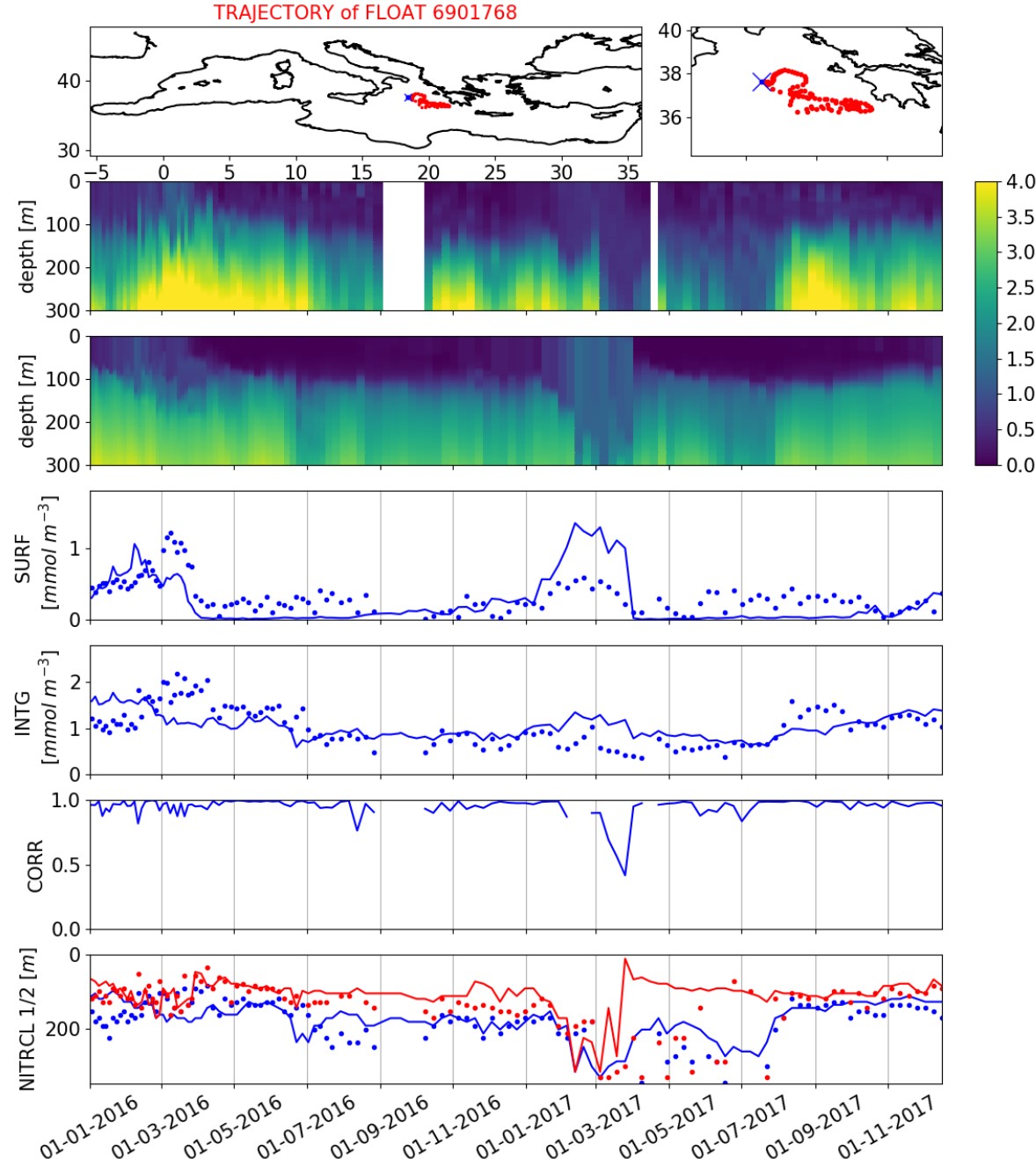

**Figure 10: Time evolution of BGC-Argo float 6901768. Top panel: trajectory of the BGC-Argo float (red dots), with deployment position (blue cross); Hovmöller diagrams of nitrate concentration (mmol/m³) from float data (2nd panel) and model outputs (3rd panel) matched-up with float position for the period 2016-2017. Computation of selected skill indexes for model (solid line) and float data (dots): nitrate concentration at surface (SURF, 4th panel) and 0-200 m vertically averaged concentration (INTG, 5th panel), correlation between vertical profiles (CORR, 6th panel), depth of the nitracline computed as NITRCL1 (blue) and NITRCL2 (red; bottom panel).**

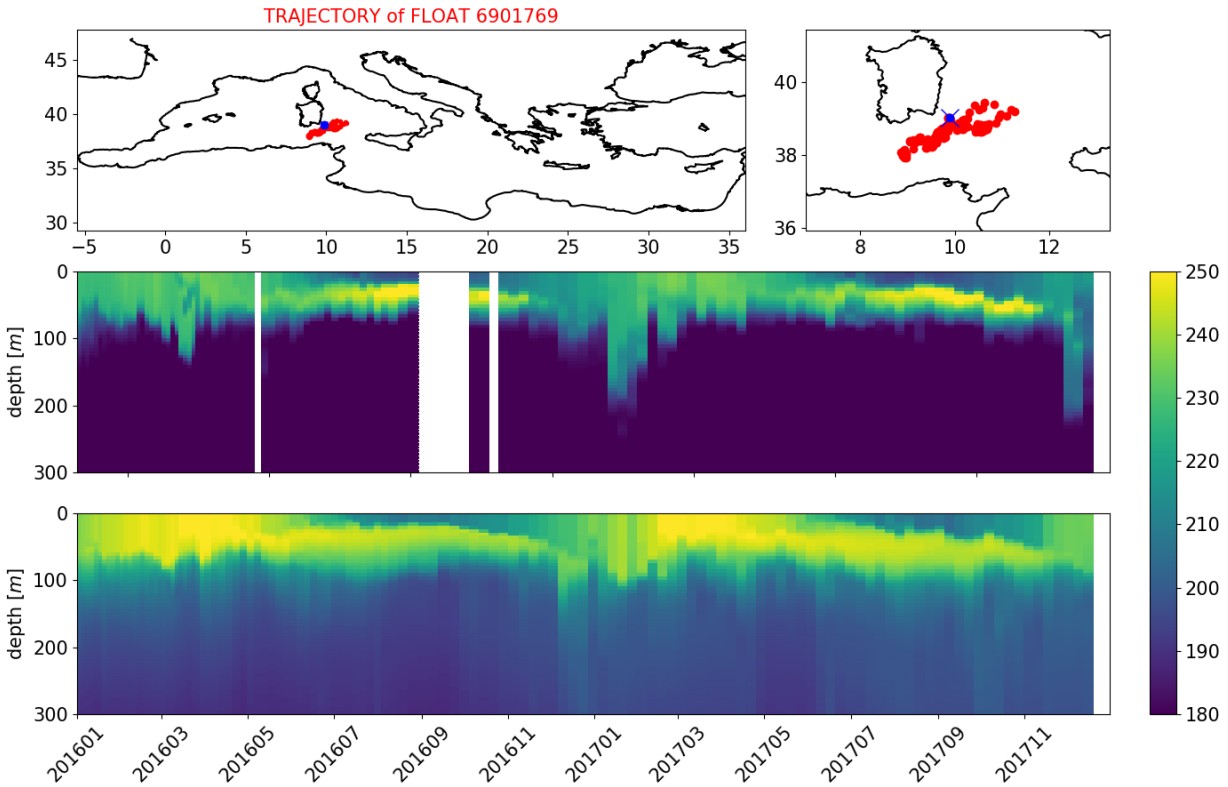

**Figure 11: Time evolution of BGC-Argo float 6901769. Top panel: trajectory of the BGC-Argo float (red dots), with deployment position (blue cross); Hovmöller diagrams of oxygen concentration (mmol/m³) of one selected BGC-Argo float 6901769 (middle panel) and model outputs (bottom panel) matched-up with float position for the period 2016-2017.**

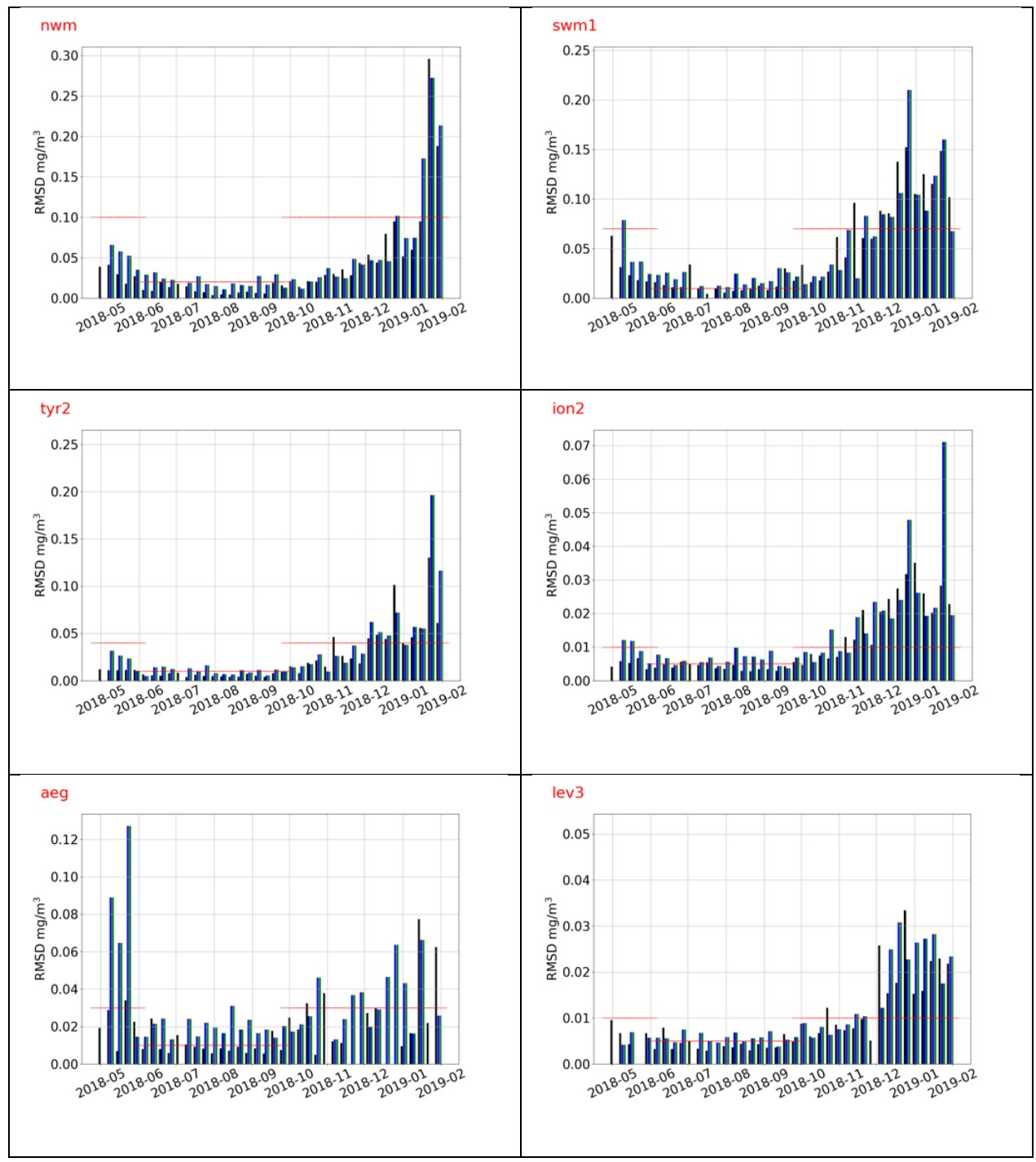

**Figure 12: Sub-basin RMSD between surface chlorophyll model forecast at lead time 24 (T1, black), 48 (T2, blue) and 72 hours (T3, green) and daily satellite maps. As benchmark reference, the two seasonal mean RMSD values computed from 2016-2017 pre-operational qualification run are shown (red lines).**

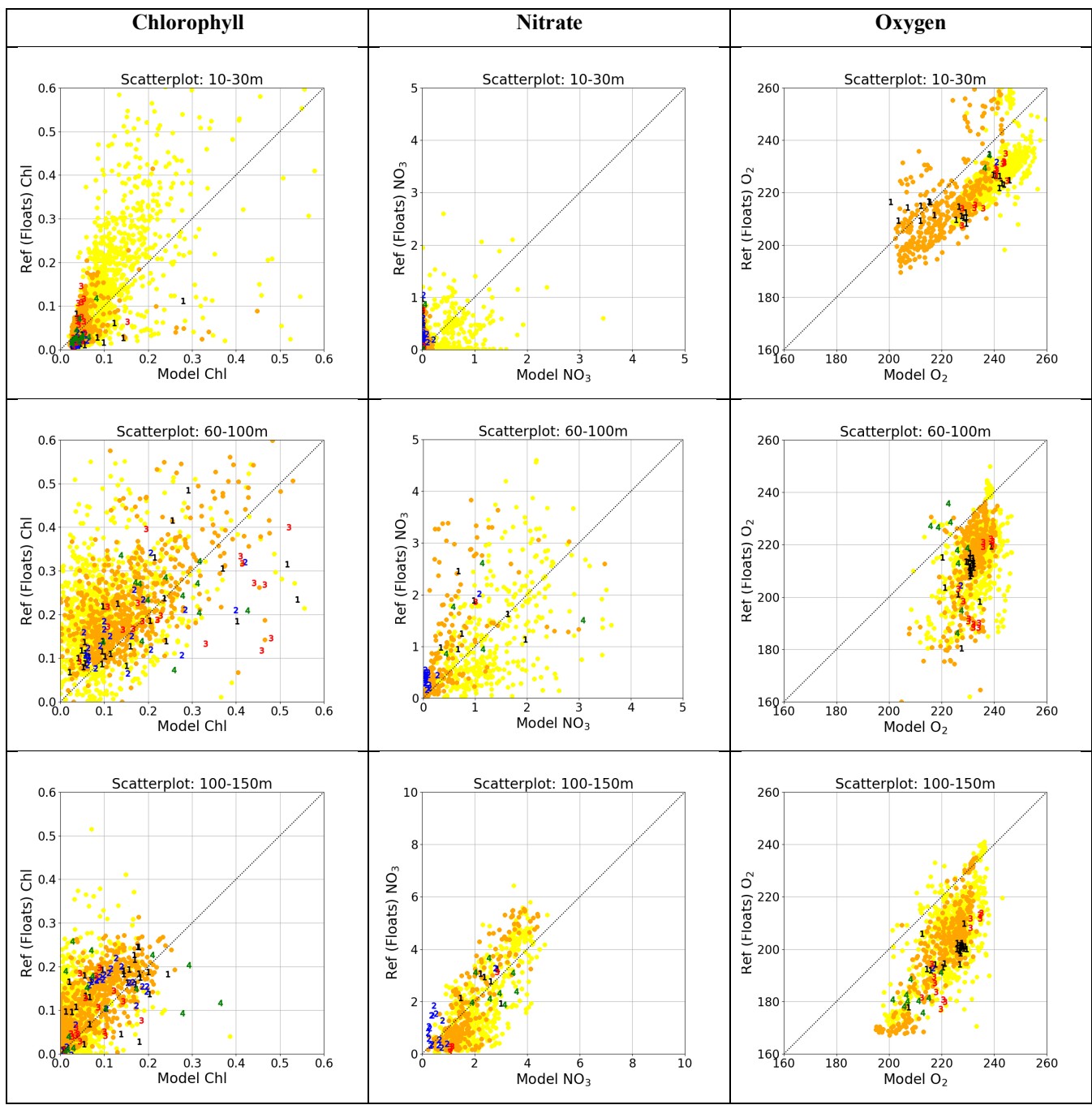

**Figure 13: Scatter plots of reference (y-axis) versus model forecast (x-axis) for chlorophyll (left column), nitrate (middle column) and oxygen (right column) at different vertical layers: 10-30 m, 60-100 m and 100-150 m. Model forecast are labelled with numbers from 1 to 4 corresponding to lead time from T1 to T4. As benchmark reference, the 2016-2017 pre-operational qualification results are shown for a selected period of investigation (May to August, orange dots) and for the other periods (yellow dots).**

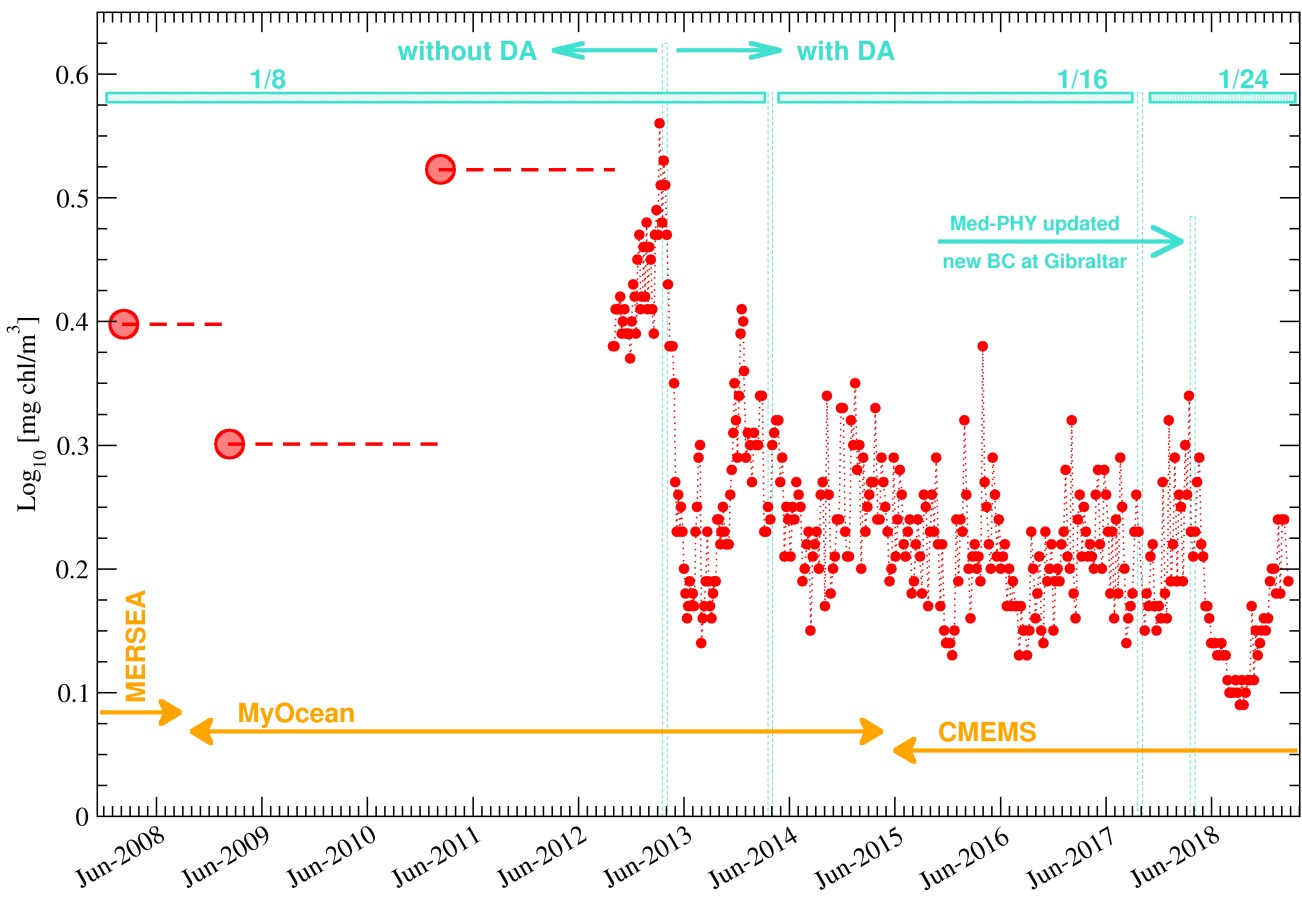

**Figure 14: RMSD of the surface chlorophyll concentration with satellite data. The use of logarithmic units has been the standard for RMSD since the implementation phase. Regular, weekly product quality assessment (red dots and dotted line) started at the end of 2012. Before, quality assessment was performed only occasionally for specific periods (e.g. Teruzzi et al., 2011; Tonani et al., 2012; large red dots) and thick red lines). In the plot, we identify the different projects (yellow), the start of data assimilation (April 2013), and the increase of horizontal resolution (cyan).**

| | n. of BGC-Argo floats active | n. of profiles |
|---|---|---|
| Chlorophyll | 28 | 2532 |
| Nitrate | 13 | 1406 |
| Oxygen | 15 | 1596 |

**Table 1: Synthesis of the BGC-Argo floats dataset for Mediterranean Sea used in the present study, for chlorophyll, nitrate and oxygen.**

| Annual mean [gC/m²/y] | Model Lazzari et al. (2012) | Satellite Bosc et al. (2004) | Satellite Colella (2006) | In situ Siokou-Frangou et al. (2010) | CMEMS MedBFM qualification run |
|---|---|---|---|---|---|
| Mediterranean Sea | 98±82 | 135.5 | 90±48 | | 127±42 |
| Alboran Sea (alb) | 274±155 | 230 | 179±116 | | 249±56 |
| South West Med –West (swm1) | 160±89 | 162 | 113±43 | | 188±22 |
| South West Med –East (swm2) | 118±70 | 162 | 102±38 | | 162±12 |
| North West Med (nwm) | 116±79 | 170 | 115±67 | 105.8-119.6; 86-232*; 140-170** | 149±18 |
| Levantine (lev1+lev2+lev3+lev4) | 76±61 | 105 | 72±21 | 59*** | 105±40 |
| Ionian Sea (ion1+ion2+ion3) | 77±58 | 120 | 79±23 | 61.8 | 107±18 |
| Tyrrhenian Sea (tyr1+tyr2) | 92±5 | 137 | 90±35 | | 139±25 |

**Table 2: Annual averages and short period estimates of the vertically integrated primary production for some selected sub-regions. Estimates are from multi-annual simulation (Lazzari et al., 2012), from satellite model (Bosc et al., 2004; Colella, 2006), from in situ estimates (Siokou-Frangou et al., 2010) and from the present CMEMS qualification run. Notes: *only DYFAMED station; **Southern Gulf of Lions; ***only Cretan Sea.**

| | SURF [mg/m³] | | INTG [mg/m³] | | | DCM [m] | | MWB [m] | | average number of available profiles per month |
|---|---|---|---|---|---|---|---|---|---|---|
| | BIAS | RMSD | BIAS | RMSD | CORR | BIAS | RMSD | BIAS | RMSD | |
| alb | -0.04 | 0.24 | -0.01 | 0.05 | 0.48 | -19 | 19 | n.c. | n.c. | 2 |
| swm | -0.02 | 0.07 | -0.07 | 0.08 | 0.70 | 4 | 15 | n.c. | n.c. | 4 |
| nwm | -0.07 | 0.18 | -0.05 | 0.07 | 0.78 | -9 | 15 | 43 | 61 | 12 |
| tyr | -0.04 | 0.07 | -0.07 | 0.08 | 0.72 | -5 | 13 | n.c. | n.c. | 10 |
| adr | 0.01 | 0.04 | -0.04 | 0.05 | 0.71 | -3 | 12 | n.c. | n.c. | 4 |
| ion | -0.03 | 0.06 | -0.03 | 0.04 | 0.85 | 7 | 14 | -29 | 56 | 21 |
| lev | -0.01 | 0.05 | -0.04 | 0.05 | 0.73 | 4 | 18 | -52 | 60 | 22 |

**Table 3: Averages of the chlorophyll indicators based on the BGC-Argo floats and model comparison for the period January 2016 – December 2017. The indicators are the BIAS and RMSD of the surface (SURF) and of the vertically 0-200 m averaged (INTG) chlorophyll concentration, the correlation between model and BGC-Argo float data (CORR), the BIAS and RMSD of the depth of**

the Deep Chlorophyll Maximum (DCM) and depth of the vertically mixed winter bloom (MWB). Statistics are computed for selected aggregated sub-basins; MWB statistics are not computed (n.c.) for some sub-basins.

| | SURF [mmol/m³] | | INTG [mmol/m³] | | CORR | NITRCL1 / NITRCL2 Mean OBS [m] | NITRCL1 [m] | | NITRCL2 [m] | | average n. of available profiles per month |
|---|---|---|---|---|---|---|---|---|---|---|---|
| | BIAS | RMSD | BIAS | RMSD | | | BIAS | RMSD | BIAS | RMSD | |
| swm | 0.13 | 0.22 | 0.04 | 0.78 | 0.92 | 100 / 106 | -7 | 27 | -23 | 38 | 1 |
| nwm | -0.12 | 0.53 | -0.61 | 1.44 | 0.98 | 64 / 96 | 35 | 39 | 2 | 20 | 2 |
| tyr | -0.01 | 0.18 | -0.68 | 0.76 | 0.97 | 82 / 82 | 8 | 17 | -5 | 19 | 4 |
| ion | -0.04 | 0.29 | -0.03 | 0.36 | 0.93 | 148 / 111 | -1 | 38 | -14 | 37 | 6 |
| lev | 0.18 | 0.26 | 0.53 | 0.63 | 0.93 | 176 / 153 | -31 | 37 | -42 | 57 | 17 |

Table 4: Averages of the nitrate indicators based on the BGC-Argo floats and model comparison for the period January 2016 – December 2017. The indicators are the correlation between model and BGC-Argo float data (CORR), the BIAS and RMSD of the surface (SURF) and of the vertically 0-200 m averaged (INTG) nitrate concentration, the BIAS and RMSD of the depth of the nitracline computed as NITRCL1 and NITRCL2. Statistics are computed for selected aggregated sub-basins. For a reference, the mean value of NITRCL1 and NITRCL2 estimated from BGC-Argo data is included (Mean OBS).

| | 0-10 m | 10-30 m | 30-60 m | 60-100 m | 100-150 m | 150-300 m | 300-600 m | 600-1000 m | average n. of available profiles |
|---|---|---|---|---|---|---|---|---|---|
| swm | 13.2 | 13.1 | 12.4 | 27.5 | 27.2 | 27.3 | 25.1 | 18.8 | 26 |
| nwm | 14.6 | 14.7 | 19.0 | 25.6 | 25.0 | 24.3 | 23.8 | 20.6 | 107 |
| tyr | 15.3 | 14.4 | 14.8 | 24.5 | 24.2 | 24.5 | 25.5 | 21.5 | 217 |
| adr | 5.8 | 12.2 | 8.1 | 5.0 | 3.9 | 3.7 | 4.5 | 10.4 | 78 |
| ion | 12.9 | 11.9 | 10.2 | 14.4 | 17.9 | 18.4 | 18.6 | 16.6 | 242 |
| lev | 13.9 | 12.5 | 11.9 | 18.0 | 21.3 | 21.2 | 25.2 | 21.7 | 388 |

Table 5: RMSD of the oxygen difference between BGC-Argo float and model at the float position and time. Statistics are computed for sub-basins and given layers, for the period January 2016 – December 2017.

| | PRE-OPERATIONAL | T1 | T2 | T3 | T4 |
|---|---|---|---|---|---|
| NO3 | 0.78 | 0.79 | 0.68 | 0.63 | 0.60 |
| O2 | 18.05 | 20.08 | 18.62 | 25.09 | 22.46 |
| CHLA | 0.13 | 0.06 | 0.05 | 0.07 | 0.08 |

Table 6: RMSD of BGC-Argo float and model comparison for the pre-operational qualification run and for the first 4 days of forecast of the Med-BIO forecast system (T1 to T4) since April 2018. Statistics are computed using the layers 0-300 m for nitrate and oxygen and 0-150 m for chlorophyll.