# Peer review of "Novel metrics based on Biogeochemical Argo data to improve the model uncertainty evaluation of the CMEMS Mediterranean marine ecosystem forecasts"

_Ocean Science, 2018_

## Referee Comment (RC1) · Anonymous Referee #1 · 8 Feb 2019

General comments:

The manuscript entitled "Marine Ecosystem forecasts: skill performance of the CMEMS Mediterranean Sea model system" by Salon et al. aims to evaluate the skill performance of the biogeochemical model for the Mediterranean component of the CMEMS biogeochemical products. They build their evaluation approach on two complementary simulations: 1) pre-operational run which consists of a 2-year re-analysis biogeochemical simulation daily coupled offline to the physical component, and 2) operational biogeochemical run as of April 2018. They perform the model evaluation by GODAE Class 1 and 4 metrics allowing them to qualitatively and quantitively document

the model consistency in reproducing the key Mediterranean biogeochemistry features and its accuracy to routinely reproduce the observed values. In addition to model data comparison to climatologies, satellite images and literature, their use of ARGO floats mounted with biogeochemical sensors for a Class 4 (specific time and location) skill performance evaluation framework is the merit of this manuscript that stands out as a novel concept, and introduces to the modelling community a tool for validation of biochemical variables that is high resolution in time, with off-shore coverage that is not limited to ocean surface and not hindered by cloud cover. The scarcity of observations is a challenge for a sound model validation and this is much more of an issue for the biogeochemical variables, thus new tools as such are of high value stressing that the floats not only allow data for a hindcast simulation, but can be used as a supplement to forecast simulations. The authors introduce techniques that are applicable to many other regional/global cases, and advancements may further benefit the modelling community, while they acknowledge that the use of floats requires further improvements such as the need for better corrections/adjustments on the raw float data in order to make a direct comparison.

In summary, authors start by introducing the modelling framework for the Mediterranean, and build the case for the skill assessment by introducing the model simulations and the dataset used for model validation, present examples for different state variables and their statistics using climatologies, satellite images, literature and float data, and finalise by discussing the outcome of the techniques introduced in this manuscript. The language was clear, and the text was easy to follow, and introduces (for the case of biogeochemical modelling) new concepts and tools.

Specific comments:

(P8 L1-10) Authors discuss further in the manuscript that direct comparison of model results with the sensor data should be done with caution as the applied corrections may not reflect true value. Thus, application of the introduced metrics which is consistent within itself (e.g. normalising the data to its own surface value) is a good approach. As these are new metrics, for their applicability to other regions, the choice of the criteria deserves a discussion. Do 10% of the surface chl value, or 2mmol/m3 nitrate have a significant meaning? Are these values applicable enough throughout the Mediterranean or have the authors seen regional inconsistencies? Are the strict choice of seasons (jan-mar and apr-oct) valid in practice (can't tell much from Figure 9, but wouldn't using MLDs from the ARGOs yield better estimation of which criteria to use? ARGO profiles show deep chl formation in late Jan and late March 2016 hence much deeper mwb than then model)

(P11 L8-10) Does this partly explain the lower modelled NO3 concentrations (e.g. nwm) due to the lack of N river load time-series? How does model perform in terms of N/P ratios? Does it represent the high N/P ratio character of the Med Sea and its regional differences?

(Fig9) How does BGC-Argo surface chl compare with the satellites? P11 L4 suggest the model has a higher (0.015) bias for the winter (model vs satellite), supported by Fig8 with more pronounced bias for the west/northwest Med, while paragraph of P11 L32 suggest the model has lower values when compared to BGC-Argo data. Is there a consistent ratio between satellite and float data, and how applicable is it to use global correction of division by 2 as suggested by Roesler et al. (2017) taking into account the regionally different ratios shown in the same article. As the Mignot et al. (2018) manuscript is in review, I cannot comment about their results but can the application of their method suggest different correction factors with a better regional fit?

(P14 L20-23) Authors point out that the introduced BGC-ARGO related metrics are already being implemented for data assimilation purposes with consequent improvements in model solutions. Before the assimilation phase (e.g. pre-opetaional runs), does the skill assessment documented here (of BGC-Argo metrics) reveal any prior messages for model parameter adjustment such as for light attenuation or nutrient assimilation rates, or errors of physical model origin? I can see the use of this dataset not only for forecasting purposes and skill assessment purposes, but its high resolution

coverage including ocean interior is of high value. A short comment on that would be good scientific addition to the manuscript.

I see that the manuscript is designed as a document for the overall skill assessment of MedBFM, but both the abstract and the manuscript throughout have stressed the importance and usefulness of their new metrics (GODAE Class 1 and 4, and especially the use of BGC-Argo), and I agree with them, and these sections of the manuscript stand out as the novel scientific content. The title fails to give this message and won't promote this novel scientific content of the manuscript. I leave it to the authors consideration.

Technical corrections:

(P2 L1) As a user-driven

(P3 L21) semi-labile

"In situ" appear as in-situ in various parts of the text. Ocean science journal asks for the use "in situ".

(P7 L16) replace relays by relies

(P7 L20) remove first "run"

(P8 L8) Add CORR acronym

(P9 L24) model simulates well

(P18 L1) represent a useful

(Fig10) Color code for NITR1 and NITR2 is missing

(Fig13) Authors used CHL for 60-100 m twice, suggesting NO3 is missing at that depth range in the figure.

---

## Referee Comment (RC2) · Anonymous Referee #2 · 14 Feb 2019

General Comments

This study deals with the assessment of an operational BGC system on the Mediterranean Sea. It is organized as follows: (1) the study of the pre-operational qualification run (2-year long) both in terms of consistency and skill performance; (2) the near-real time validation of the operational run (from April 2018 to November 2018) in terms of skill performance. This manuscript is well structured and gives an overview of the overall quality of the BGC simulation using classical CLASS 1 metrics but also innovative CLASS4 metrics (using quality-controlled BGC-Argo data). The authors use a BGC-Argo database consisting of 28 floats with chlorophyll, 13 with nitrate and 15 with

oxygen. The weaknesses of this study are: (1) It doesn't give an overview of other pub-
lished simulations on the Mediterranean Sea. The reader is not able to appreciate the
quality of this simulation in relation to other BGC models in the literature; (2) It doesn't
manage to rationalize the discrepancies diagnosed by the comparison with observa-
tions and to disentangle the origin of the observed biases (problems due to the physical
model, the biogeochemical parameterizations or to the sensor quality control?). They
are a lot of metrics that are not enough used to suggest some solutions and to improve
the future systems. Some figures need also to be improved.

Specific comments

P3 L26-27: "surface chlorophyll . . . of CMEMS" Which CMEMS product? Is it the same
as the one used for assessment on Fig 3 and Fig 4?

P4 L10: Which database do you use for river inputs? Reference?

P4 L15-16: "Additional 2D fields include the surface data for solar irradiance and wind
stress": Where are these data from?

P5 L11: Is the spin-up forced by a climatological year?

P6 L16-20: is it the same dataset as the one used in the data assimilation scheme?
Can you precise the temporal frequency. Daily?

P6 L24-30: do you use the BGC-Argo dataset available on Coriolis website?

P11 L19: prefer 'averaged' instead of 'integrated'

P11 L20 and Fig9: why do the correlation values vary so abruptly and reach zero
sometimes in winter?

P11 L22-23 and Fig9: This sentence is not convincing. Why these differences between
modelled and observed depth of MWB? What is the difference between the mixed layer
depth (MLD) and the depth of MWB? Why did you choose this index? Did you compare
the mixed layer depths along the float trajectory?

**OSD**

Fig9: in summer 2017, there is no diagnosed DCM for model. Why?

Fig 10: skill index 4th row: the legend is missing for NITRCL1/NITRCL2. Why is the blue index missing for model in the beginning of 2017?

Fig10-11: Authors don't show depths below 300m. For oxygen, the colorbar is saturated below 200 umol/L. It doesn't allow to study the nitrate gradient and the OMZs.

P12 L21: I would say 'solubility' instead of 'saturation'

P12 L22: What are the consumption terms?

Fig 12: needs to be improved (quality). Do you compare to the daily NRT L3 chlorophyll product? The figure seems to be at a coarser temporal resolution. Moreover, the yaxis labels don't appear correctly (they are truncated). I don't understand why the red curve is constant during the summer.

P13 L15: 0.041 instead of 0.41 mg/m3?

P13 L18-20 and Fig 13: Authors don't comment much the Fig13. For example, the nitrate dots are very scattered (between 0 and 100m depth), do you think that it is due to nitrate sensor anomalous values or is it a problem in the physical or biological models? Oxygen panels display a bias (model overestimates float data): do you think it is a bias in the model or in the data?

Table 6: I am wondering why the chlorophyll RMSD for the pre-operational is twice as large as the one for the T0 forecast. Do you have an explanation for the nitrate RMSD decrease from T0 to T3?

Technical comments:

P4 L16-17: "are used respectively ...module". I think you should switch both: "as input for the BFM optical module and to solve the gas air-sea exchanges"

P4 L21: temperature AND salinity

[Figure]

P6 L14: the same AS

P7 L2: SchmeCHtig

P7 L16: relies instead of relays

P9 L6-7: surface bloomS in nwm appear (remove the s)

P9 L9-10: NODC-OGS instead of OGS-NODC for more consistency with the beginning of the manuscript

P10 L7: "a large seasonal cycles": remove the "s" in cycles

P11 L10 "river plume" : remove the "s"

P18 L1: represent a (remove the 'n')

Fig 14: in caption, 'increase' instead of 'increment'?

---

## Author Comment (AC1) · 8 Mar 2019

We thank the reviewer for her/his positive *comments*. We report here our proposed replies to the specific points. If accepted, these notes will be properly included in the revised version.

*1. (P8 L1-10) Authors discuss further in the manuscript that direct comparison of model results with the sensor data should be done with caution as the applied corrections may not reflect true value. Thus, application of the introduced metrics which is consistent within itself (e.g. normalising the data to its own surface value) is a good approach. As these are new metrics, for their applicability to other regions, the choice of the criteria deserves a discussion. Do 10% of the surface chl value, or 2mmol/m3 nitrate have a significant meaning? Are these values applicable enough throughout the Mediterranean or have the authors seen regional inconsistencies? Are the strict choice of seasons (jan-mar and apr-oct) valid in practice (can't tell much from Figure 9, but wouldn't using MLDs from the ARGOs yield better estimation of which criteria to use? ARGO profiles show deep chl formation in late Jan and late March 2016 hence much deeper mwb than then model)*

REPLY 1 - The metrics concerning chlorophyll and the relative two periods for the computation have been derived from the outcomes of Lavigne et al. (2015), who identified some standard shapes for chlorophyll profiles from the analysis of a large number of chlorophyll (fluorescence data) profiles in the Mediterranean Sea (see their Fig. 2). In particular, our summer period is based on the consideration that the DCM shape profile (useful to define our DCM index) is typically observed from April to October (Fig. 5 by Lavigne et al., 2015). Then, three other shape profiles (i.e., "homogeneous", "high surface chlorophyll-HSC" and "complex" in Fig. 2 by Lavigne et al., 2015) are characterized by decreasing values with depth, and typically occur between January and March in different Mediterranean regions (Fig. 5 by Lavigne et al., 2015).
The choice of the 10% criterion for the MWB index was made after a sensitivity analysis varying the limit between 1 to 10% (not shown). The value of 10% gave results qualitatively consistent with those reported by Lavigne et al. (2015). Further, lower percentage values gave more unclear patterns because the depth increases substantially and the thickness of model layers has also an impact (i.e. thickness of model layers is 5 m at 70 m depth, 7 m at 100 m and 10 m at 140 m). Thus, the selection of the period might depend on a preliminary investigation of the typical behavior of the ecosystem. Alternatively, an analysis employing machine learning technique and additional data such as DCM values can provide a better tool to identify shape profiles and their distribution in time, however this is out the scope the present work and might require a considerable amount of data.
The use of the MLD for the choice of seasons' limits represents a promising alternative option, since it would account for the specific conditions at each float profile. However, it must be noted that there are several possible definitions of MLD which can give slightly different results, thus a sensitivity analysis of the biogeochemical metrics to the MLD definition would be necessary to tune the choice of the criteria based on MLD. Therefore, we decided to adopt an a priori, while rigid, temporal subdivision based on literature to test the feasibility of the computation of the chlorophyll criteria.

Regarding the criteria for nutricline metrics, we are aware that selecting an unique criterion to detect the nutricline might be sensitive and controversial. Further, we think that the important aspect is to rather track the time evolution of the nutricline. For these reasons, we tested two different approaches: the depth of the 2 mmol/m$^3$ concentration isopleth (NITRCL1) and the depth of the maximum nitrate vertical gradient (NITRCL2). According to Manca et al. (2004), the values of nitrate concentration at depth higher than 400 m are around 4-5 in the eastern basin and 6-7

mmol/m$^3$ in the western, therefore the isopleth of 2 mmol/m$^3$ can be considered a safe value to detect the rapid change between the very low concentration typically measured at the surface and the high concentration at depth in all areas of the Mediterranean.

*2. (P11 L8-10) Does this partly explain the lower modelled NO3 concentrations (e.g. nwm) due to the lack of N river load time-series? How does model perform in terms of N/P ratios? Does it represent the high N/P ratio character of the Med Sea and its regional differences?*

REPLY 2 - We agree: the effect of the lack of high frequency data of nutrient discharges is one of the most important sources of uncertainty at the daily/weekly time scale (not at seasonal/annual scale) and at very local coastal scales, as discussed by Teruzzi et al. (2018). Indeed we highlighted this potential issue at (P17 L13-17).
Uncertainty in NWM is partly related to a possible underestimation of the river input forcing and possibly to the effect of lateral circulation from ALB and SWM1 surface waters (see Fig. 5 in the manuscript). A sensitivity analysis of the impacts of the different factors would help in elucidating the most relevant factor. However, the relevant point is that the fine sub-division of the Mediterranean Sea in 16 sub-basins allows to detect the relevant spatial gradients and, thus, to highlight possible issues. The choice of 16 sub-basins was a trade-off between having a number of areas as larger as possible, the need for having robust in situ statistics and the known characteristic on dynamics derived from literature. Thus, our proposed sub-division is a relevant result in itself. We propose to include this point in the part concerning the discussion on the sub-basin subdivision at (P16 L10-17), adding a comment on the detection of the surface nitrate underestimation of Fig. 5 at (P16 L17):
"In fact, the comparison of nutrients profiles of Fig. 5 allows to highlights the satisfactory model performance in reproducing the mean spatial gradients and possible anomalies such as the underestimation of Nitrate in NWM sub-basin, which is related partly to possible underestimation of the terrestrial input and partly to the impact of the incoming Atlantic waters".

Concerning N/P ratios, the performance of OGSTM-BFM model system was assessed in Lazzari et al. (2016) for nitrate and phosphate separately, showing a general higher-than-Redfield ratio in the Mediterranean Sea (closer to a N:P=22 ratio, with exception of the Alboran Sea area, characterized by a lower-than-Redfield ratio) and significant spatial variability. The present operational configuration of the MedBFM incorporates the results presented by Lazzari et al. (2016; see their Fig. 7 here reported as Fig.R1, for further details please refer to the paper) and provides consistent results on N:P ratio (see also N and P values in Fig. 5).

[Figure]

Fig.R1 - Results from Lazzari et al. (2016), model-derived vertical profiles of phosphate (x-axis) and nitrate (y-axis) averaged by sub-basins (reported in Fig. 1 of Lazzari et al., 2016) for the period 1999–2004. Each entry represents the spatial-temporal average at a certain depth: 0–50 m layer (circles) and 50-bottom layer (triangles).

*3. (Fig9) How does BGC-Argo surface chl compare with the satellites? P11 L4 suggest the model has a higher (0.015) bias for the winter (model vs satellite), supported by Fig8 with more pronounced bias for the west/northwest Med, while paragraph of P11 L32 suggest the model has lower values when compared to BGC-Argo data. Is there a consistent ratio between satellite and float data, and how applicable is it to use global correction of division by 2 as suggested by Roesler et al. (2017) taking into account the regionally different ratios shown in the same article. As the Mignot et al. (2018) manuscript is in review, I cannot comment about their results but can the application of their method suggest different correction factors with a better regional fit?*

REPLY 3 – We thank the reviewer for having raised this point, since the inconsistency in surface chlorophyll observations between satellite and BGC-Argo floats has been already observed in our investigations (see Fig.R2) and represents a potential issue, not only for validation purposes as discussed here, but also for multi-platform data assimilation (e.g., combined assimilation of chlorophyll from BGC-Argo floats and satellite). Regional corrections of BGC-Argo float data can be advisable, such as regional algorithms for the ocean color exist for satellite, to provide chlorophyll in different regions of the global ocean. Methods such as that of Mignot et al. (2018) can be helpful proven the availability of a sufficient amount of in situ independent data.
Such investigation is off-topic with respect manuscript, we can comment that operational systems are optimal tools to test the consistency of many different sources of information as already reported in the submitted manuscript citing She et al. (2016) at P2-L23 and P17-L2.

[Figure]

Fig. R2 - Chlorophyll concentration comparison between surface BGC-Argo floats data and satellite data (CMEMS ESA-CCI Mediterranean data). Data of all floats available between 2015-2017 are here included. The matchup between data is performed on a daily basis and using a bilinear spatial interpolation. Positive deviation of BGC-Argo floats w.r.t. satellite data is evident in winter while negative deviation of BGC-Argo data is present in summer and spring months.

*4. (P14 L20-23) Authors point out that the introduced BGC-ARGO related metrics are already being implemented for data assimilation purposes with consequent improvements in model solutions. Before the assimilation phase (e.g. pre-opetaional runs), does the skill assessment documented here (of BGC-Argo metrics) reveal any prior messages for model parameter adjustment such as for light attenuation or nutrient assimilation rates, or errors of physical model origin? I can see the use of this dataset not only for forecasting purposes and skill assessment purposes, but its high resolution coverage including ocean interior is of high value. A short comment on that would be good scientific addition to the manuscript.*

REPLY 4 – We thank the review for this comment. The integration of BGC-Argo floats within an analysis and forecasting system (in terms of data assimilation) paved the way to an in-deep study of the interior of the sea and its dynamics. Given the fact that the BGC-Argo provide also profiles of PAR, salinity and temperature, which are forcing mechanisms of the biogeochemical processes (or proxies for the forcing impacting the biogeochemistry), a global analysis of the uncertainty can be made comprehensively using multivariate statistical analysis (e.g. PCA, neural network methods), with the aim of disentangling the sources of error on profiles.
However, it must be noted that this analysis would need not simple measures of distance between observations and model values (such as BIAS and RMSD), but indexes that can put in relation the shape and intensity of the profiles with the underlying processes. We think that our work provides a first step to identify and quantify several functional indexes. Another critical point is the availability of a sufficient amount of profiles for variables like nitrate and oxygen, which may allow for statistically significant analysis. A short comment will be further added on this point in the discussion section.

*5. I see that the manuscript is designed as a document for the overall skill assessment of MedBFM, but both the abstract and the manuscript throughout have stressed the importance and usefulness of their new metrics (GODAE Class 1 and 4, and especially the use of BGC-Argo), and I agree with them, and these sections of the manuscript stand out as the novel scientific content. The title fails*

*to give this message and won't promote this novel scientific content of the manuscript. I leave it to the authors consideration.*

REPLY 5 – We agree: the title will be changed highlighting the novel metrics and the usefulness of BGC-Argo float data. Possible title may become: "Skill performance of the CMEMS Mediterranean marine ecosystem forecasts: improving model uncertainty assessment using BGC-Argo floats data-based metrics".

Minor/technical comments will be also thoroughly addressed in the review, correcting in particular Figs. 10 and 13.

Best Regards

---

## Author Comment (AC2) · 11 Mar 2019

We thank the reviewer for her/his *comments*. We report here our proposed replies to the two major weaknesses and the specific comments: if accepted, they will be properly included in the revised version of the manuscript.

*(1) It doesn't give an overview of other published simulations on the Mediterranean Sea. The reader is not able to appreciate the quality of this simulation in relation to other BGC models in the literature;*

REPLY 1 – We agree. The revised version will refer to other biogeochemical model applications in Mediterranean Sea. In particular, considered the focus of the manuscript (i.e., biogeochemical operational forecast and its uncertainty estimates), we will include the Poseidon operational system (HYBRID-POM-ERSEM model, Tsiaras et al., 2017; Petihakis et al., 2018). Besides that, there are other recent publications describing multi-annual simulations (Mattia et al., 2013; Macias et al., 2014; Guyennon et al., 2015; Richon et al., 2017) but we think it is out of the scope of this manuscript to list all the models and to provide an overview of their characteristics. Nevertheless, one important aspect that we would like to highlight in the introduction is an overview of the data availability for validating biogeochemical simulations in the Mediterranean Sea. In particular, the most used variable for validation is satellite-derived surface chlorophyll (Tsiaras et al., 2017, for year 2000; Mattia et al., 2013; Macias et al., 2014; Guyennon et al., 2015; Richon et al., 2017, for a portion or the whole respective investigated multi-year period). In situ measurements from vessels and scientific cruises are also used in Richon et al. (2017) and Guyennon et al. (2015), but allow only to validate very limited time and space portions of the simulations (fixed stations in time or single transects in a very limited time range). Only a few basin-wide validation frameworks, especially for nutrients, are based on comparison with climatology, e.g. Tsiaras et al. (2017) use a seasonally aggregated reference for the whole Mediterranean Sea build on 1990-1999 data from SeaDataNet. Very rarely the vertical proprieties of biogeochemistry are assessed (e.g. Guyennon et al., 2015 and Teruzzi et al., 2014).
Our work introduces the use of BGC-Argo floats data as observational counterpart for the model validation, and presents a comprehensive validation framework from basin-wide and seasonal scale to mesoscale and weekly scale (following GODAE recommendations), with an accuracy level depending on the specific variable and the availability of reference data: for this reason, it may represent a novelty.

*(2) It doesn't manage to rationalize the discrepancies diagnosed by the comparison with observations and to disentangle the origin of the observed biases (problems due to the physical model, the biogeochemical parameterizations or to the sensor quality control?). They are a lot of metrics that are not enough used to suggest some solutions and to improve the future systems*

REPLY 2 – We thank the reviewer for this comment. We will enrich the comments related to the results gathered from the metrics analysis and coherently relate them with our guesses on possible causes. However, it must be pointed out than the discussion in the submitted manuscript already outlines the general weaknesses of the MedBFM model, which will require future development work (as planned within Med-MFC and requested by the CMEMS "continuous improvement" paradigm). Nevertheless, if accepted, in the revised version we will improve our analysis by reporting and rationalizing the cases when our validation framework can detect specific discrepancies.

For example, as reported in the reply to RC1 (point 2), we highlight that our definition of 16 sub-basins revealed to be a sensible choice since it allowed to demonstrate the model good performance in reproducing the mean spatial gradients within the domain (Fig. 5 of the submitted manuscript) and possible anomalies, such as the underestimation of nitrate in the NWM sub-basin in the subsurface layer that is related partly to possible underestimation of the terrestrial input and partly to the impact of the incoming Atlantic waters.

Another possible detected discrepancy that we would like to mention is the presence of some overestimation for the oxygen field between model and BGC-Argo profiles below 200 meter. However, since the quality check protocol of oxygen data from BGC-Argo floats is continuously progressing we envisage the need for a specific study regarding the oxygen validation and the analysis of oxygen variability simulated by the MedBFM model (see also point 12 of the specific comments).

Further (as also reported in the reply to RC1, point 4), the metrics based on BGC-Argo data (e.g., DCM and MWB metrics) are very innovative and informative. BGC-Argo floats provide simultaneous measurements of physical variables, such as profiles of PAR, salinity and temperature, that act as forcing of the biogeochemistry processes (or proxies for the forcing impacting the biogeochemistry). Thus, an integrate analysis of the model-observation uncertainty can be made comprehensively using multivariate analysis (e.g. PCA and neural network methods), with the aim to disentangle the error sources on vertical profiles. Moreover, we have to note that such analysis would need not just simple measures of distance between observations and model values (such as BIAS and RMSD), but indexes that can put in relation the shape and intensity of the biogeochemical profiles with the underlying processes (light limitation, temperature kinetic dependencies, nutrient availability through vertical fluxes, vertical dispersion through mixing). We think that our work (and the proposed metrics) provide a first step towards the identification and quantification of several functional indexes. Another critical point is the availability of a sufficient amount of float profiles for variables like nitrate and oxygen, which would be required for a statistically significant analysis capable to cover most of the Mediterranean regions and a longest period of the year. A comment on this issue will be added as well.

Specific comments:

*1. P3 L26-27: "surface chlorophyll . . . of CMEMS" Which CMEMS product? Is it the same as the one used for assessment on Fig 3 and Fig 4?*

REPLY 1 –The product is the OCEANCOLOUR_MED_CHL_L3_NRT_OBSERVATIONS_009_040 as described in section 2.2 ("Set up of the pre-operational qualification simulation for Med-BIO") at (P5, L7-8) and section 3 ("Reference datasets for validation") at (P6, L17-20).

Then, it is important to remind (as explained at P7, L24-27) that this product is used in the data assimilation scheme (at the beginning of each weekly run at day T-7) and then to assess the skill of the forecasts (as in Fig. 3 and 4 for the pre-operational run, and in Fig. 12 for the forecast run) before it is used for assimilation at the next weekly run. From Cossarini et al. (2019), following Mattern et al. (2018), the root mean square (RMS) of the differences between observation and prior model solution provides a measurement of the data assimilation performance and represents a short-term forecast skill metric since it is based on observations that are going to be assimilated in the following cycles. We think that this concept is already clearly depicted in the text. However, we are open to add details if the reviewer suggests they are needed.

*2. P4 L10: Which database do you use for river inputs? Reference?*

REPLY 2 – Details on the boundary conditions for river nutrients inputs are reported in section 2.2, 5th and 6th items (P5, L17-26), and derive from the dataset built during the PERSEUS FP7 project. We will also include that the dataset is based on a total of 39 rivers with runoff larger than 50 $m^3$/s. To the best of our knowledge, this deliverable represents the most up-to-date information about terrestrial nutrient discharges for the Mediterranean basin.

*3. P4 L15-16: "Additional 2D fields include the surface data for solar irradiance and wind stress": Where are these data from?*

REPLY 3 – These data are provided via the offline coupling with MED-PHY component, which derives the atmospheric forcing from the 6-hours ECMWF operational analysis and forecast fields at 0.125° horizontal-resolution. For details please refer to CMEMS catalogue: http://marine.copernicus.eu/services-portfolio/access-to-products/?option=com_csw&view=details&product_id=MEDSEA_ANALYSIS_FORECAST_PHY_006_013.
The source of the 2D fields will be added to the revised text.

*4. P5 L11: Is the spin-up forced by a climatological year?*

REPLY 4 – The spin-up is based on year 2016 run without data assimilation. This procedure is coherent with that adopted for the spin-up of the pre-operational run of the physical component of the Med-MFC (see details in Clementi et al., 2018).

*5. P6 L16-20: is it the same dataset as the one used in the data assimilation scheme? Can you precise the temporal frequency. Daily?*

REPLY 5 – Yes, the dataset is the same, and it is used at daily frequency. We will add "daily" at line P6-L17 to better clarify this point. As discussed at previous Reply point n. 1, validation is based on satellite surface chlorophyll data that are not yet assimilated.

*6. P6 L24-30: do you use the BGC-Argo dataset available on Coriolis website?*

REPLY 6 – We download our data from LOV database due to specific quality control developed in the frame of the CMEMS Service Evolution project MASSIMILI (see details at https://www.mercator-ocean.fr/en/portfolio/massimili-2/). The use of BGC-Argo floats data from Coriolis and CMEMS INSITU-TAC is presently under consideration.

*7. P11 L19: prefer 'averaged' instead of 'integrated'*

REPLY 7 – We agree with the suggestion, we will substitute it.

*8. P11 L20 and Fig9: why do the correlation values vary so abruptly and reach zero sometimes in winter?*

REPLY 8 – We thank the reviewer for having raised this issue. The plots of Fig. 9 (right panels) were erroneously computed also for dates without float data (see also Reply point n. 10, and revised Fig.9 in Fig.R1). In other few cases, very low correlation values were related to some possible inconsistencies in the float measurements (see for example the isolated chlorophyll values up to 0.20 mg/m$^3$ below 200 m in April 2016 and in May 2017). Our operational check of the goodness of the input data is still progressing. Thus, in these cases, we safely decided to remove from the computation these profiles. The revised version of Fig. 9 (here reported) has been corrected.

*9. P11 L22-23 and Fig9: This sentence is not convincing. Why these differences between modelled and observed depth of MWB? What is the difference between the mixed layer depth (MLD) and the depth of MWB? Why did you choose this index? Did you compare the mixed layer depths along the float trajectory?*

REPLY 9 – We introduced the MWB in order to identify an index quantifying the thickness of the layer affected by the surface chlorophyll winter bloom. As commented in the Reply to major point n. 2, this is a first step towards a specific metric to keep track of biogeochemical processes and their relationship with physical drivers.
The definition of MWB is based on the paper by Lavigne et al. (2015), who identified some standard shapes for chlorophyll profiles from the analysis of a large number of chlorophyll (fluorescence data) profiles in the Mediterranean Sea (see their Fig. 2). Three of those shape profiles (i.e., "homogeneous", "HSC" and "complex") are characterized by decreasing-with-depth values and are typically observed during the winter months in different Mediterranean regions (Fig. 5 by Lavigne et al., 2015). They are not necessarily limited by the MLD since other factors (e.g., light and nutrient conditions) play a role in this surface bloom dynamics.
Thus, the MWB index aims at detecting the thickness of the surface productive layer during winter. The specific choice of the limit of 10% was made after a sensitivity analysis varying the limit between 1 to 10% (not shown). The value of 10% gave results qualitatively consistent with those reported by Lavigne et al. (2015), whilst lower percentage values gave more unclear patterns because the depth increases substantially and the thickness of model layers has also an impact (i.e. thickness of model layers is 5 m at 70 m depth, 7 m at 100 m and 10 m at 140 m). Further, the MWB metric fails to capture the "modified DCM" shape profile (as defined in Lavigne et al., 2015) which occurred for float 6901653 during winter 2016 (Fig. 9). Thus, given the constraint in the definition and the vertical discretization of the model, the application of this index to the floats data may originate some inconsistency (as shown in Fig. 9) and under- or overestimations and uncertainty of a few decameters (see Tab. 3). Despite these limitations, we think that the MWB represents a feasible and informative metric to be coupled with the DCM metrics.
In the revised manuscript, we will explain better the scientific rationale of the MWB metrics and we will modify the original sentence ("The depth of MWB shows some inconsistency between model and float data, since it is not always computable from BGC-Argo floats data.") according to this comment.

*10. Fig9: in summer 2017, there is no diagnosed DCM for model. Why?*

REPLY 10 – We thank the reviewer: there was an error in the plot (see also Reply point n. 8). The revised version (Fig. R1) reports a wider y-axis and we observe that the DCM in 2017 appears shallower than in 2016 (please also consider the float trajectory – in the left panes of Fig. 9 –, which covers the south-western Mediterranean from Balearic Islands to Alboran Sea).

[Figure]

Fig. R1 – Revised right panels of Fig. 9: computation of selected skill indexes (1st to 4th row) for model (solid line) and float data (dots). The skill indexes are: surface (SURF) and 0-200 m vertically averaged (INTG) chlorophyll, correlation (CORR), depth of the deep chlorophyll maximum (DCM, blue) and depth of the mixed layer bloom in winter (MWB, red).

*11. Fig 10: skill index 4th row: the legend is missing for NITRCL1/NITRCL2. Why is the blue index missing for model in the beginning of 2017?*

REPLY 11 - We thank the reviewer: we will correct the caption (NITRCL1 is in blue and NITRCL2 is in red). We have also extended the maximum depth to 300 m (new version of Fig. 10, right panel is here reported as Fig.R2 for clarity). It can be observed that in the period April-July 2017 one to the two nitracline indexes (i.e., NITRCL2) computed on model output is much shallower than the one estimated from the BGC-Argo float data. Thus, the use of two indexes may help in disentangling

between errors due to model behaviour or to sensitivity of the index calculation. If accepted, a comment will be added in the revised version.

[Figure]

Fig. R2 - Revised right panels of Fig. 10: computation of selected skill indexes (1st to 4th row) for model (solid line) and float data (dots). The skill indexes are: nitrate concentration at surface (SURF) and 0-200 m vertically averaged concentration (INTG), correlation between profiles (CORR), depth of the nitracline computed as NITRCL1 (blue) and NITRCL2 (red).

*12. Fig10-11: Authors don't show depths below 300m. For oxygen, the colorbar is saturated below 200 umol/L. It doesn't allow to study the nitrate gradient and the OMZs.*

REPLY 12 – The model comparison for depths below 300 m is shown in Fig. 5 for nutrients and oxygen on 6 selected sub-basins. Given the slow dynamics of the deep layers, the comparison based on climatology demonstrates the capability of the model to reproduce the nutrient values along the Mediterranean Sea gradient and the relative minimum of the oxygen concentration below 400 m.

Regarding oxygen, the depth of the relative minimum of oxygen displayed in Fig. 5 is consistent with published information: according to Tanhua et al. (2013; see their Fig. 6), the oxygen minimum layer (OML) core in Mediterranean is located at 500 - 700 m in the eastern basin, "well below the layer of maximum S occupied by the LIW"(see sub-basins ion2, ion3 and lev4 in Fig. 5), whilst in the western basin it is shallower at around 400 m depth (see sub-basin nwm in Fig. 5, and other sub-basins are in Cossarini et al., 2018).

On the other hand, BGC-Argo floats allow to observe relatively fast dynamics (i.e., at weekly time scale according to their current 5-day sampling frequency) which are relevant for the biogeochemical dynamics in the upper layer. Thus, we used BGC-Argo floats data to verify the model capability to reproduce the mesoscale and vertical dynamics in the upper layer at the weekly time scale. This represents a very rigorous comparison, which allowed to demonstrate the timing and intensity of the evolution of signals that track biogeochemical processes in the upper layer. Therefore, we think that it would be not enough illustrative to compare and show deep profiles for model and float, also considering the ongoing and progressing advancements of the product quality practice for BGC-Argo floats data. In fact, only very recently a new product quality procedure (following Bittig et al., 2018 and Thierry et al., 2018) has been started to be implemented for oxygen to correct biases on sensor, but up to now it is not yet available for all floats in the Mediterranean.

Thus, we prefer to keep the maximum depth of Figs. 10 and 11 to 300 m to better exploit the model-float visual comparison in the upper layer (roughly identified with the euphotic layer, generally not deeper than 200 m), where most of the biogeochemical processes occur. We propose to revise the colorbar of Fig. 11 starting from oxygen value at 180 mmol/m$^3$: the new Figure 11 (see Fig.R3) displays some discrepancies (i.e., a bias of about 10-20 mmol/m$^3$) that were already commented along with Table 5. However, given the aforementioned ongoing quality improvements in the BGC-Argo float oxygen data, a formal quantification of the discrepancy between model and floats (as done for the chlorophyll and nitrate metrics) has not been defined yet, and we proposed only qualitative considerations, as reported at (P12, L10-12). In the revised manuscript, we will better clarify the limitations of the BGC-Argo data use for oxygen.

Finally, we would like to point out that with improving oxygen data quality, the use of BGC-Argo floats to investigate quantitatively the simulated oxygen will become very promising for evaluating surface and deeper layer dynamics.

A comment on the agreement with historical data (Tanhua et al., 2013) and on the improvement of the oxygen validation framework will be added in the discussion regarding future developments.

[Figure]

Fig. R3 - Revised Fig. 10: Hovmoller diagrams of oxygen concentration (mmol/m$^3$) of one selected BGC-Argo float 6901769 (top panels) and model outputs (bottom panels) matched-up with float position for the period 2016-2017.

*13. P12 L21: I would say 'solubility' instead of 'saturation'*

REPLY 13 –Thanks, the term "solubility" will be used in the revised version.

*14. P12 L22: What are the consumption terms?*

REPLY 14 – Comsumption terms are defined as respiration terms by bacteria and plankton community (4 phytoplankton and 4 zooplankton groups).

*15. Fig 12: needs to be improved (quality). Do you compare to the daily NRT L3 chlorophyll product? The figure seems to be at a coarser temporal resolution. Moreover, the yaxis labels don't appear correctly (they are truncated). I don't understand why the red curve is constant during the summer.*

REPLY 15 – We agree about the poor quality of Fig.12: we will provide an improved version of it in the revision (see a proposition of a revised panel of Fig. 12 in Fig.R4). The comparison is made, for each week, between the first 3 days of forecast (black T1, blue for T2 and green for T3) and the corresponding daily NRT L3 satellite product (as written at P13-L10). The red line represents the seasonal benchmark defined as the mean RMSD shown in Fig.8. Thus, the 2 seasonal benchmarks should inform on the average quality of forecast for the 2 periods.

[Figure]

Fig. R4 - Example of new panel of Figure 12 for sub-basin ion3. The new caption is "Sub-basin RMSD between surface chlorophyll model forecast at lead time 24 (T1, black dots), 48 (T2, blue triangles) and 72 hours (T3, green squares) and daily satellite maps. As benchmark reference, the two seasonal mean RMSD values computed from 2016-2017 pre-operational run are shown (red line)".

*16. P13 L15: 0.041 instead of 0.41 mg/m3?*

REPLY 16 – Thanks: yes, it is 0.041 mg/m3.

*17. P13 L18-20 and Fig 13: Authors don't comment much the Fig13. For example, the nitrate dots are very scattered (between 0 and 100m depth), do you think that it is due to nitrate sensor anomalous values or is it a problem in the physical or biological models? Oxygen panels display a bias (model overestimates float data): do you think it is a bias in the model or in the data?*

REPLY 17 – We acknowledge the reviewer for this point: in the submitted version, the comment to Fig. 13 is at (P13, L18-L28) but we can enrich it further. As we pointed out at (P16, L27-28), and also thoroughly commented at Reply point n. 12, the systematic bias for oxygen at depth can be either due to model uncertainty and data. Only very recently, new product quality procedure (following Bittig et al., 2018 and Thierry et al., 2018) has been starting to be implemented but up to now not available for all floats in the Mediterranean.

On the other hand, concentrations of oxygen lower than 180 mmol/m$^3$ in subsurface layer (1-150 m) appear quite anomalous for the Mediterranean Sea (see for example Tanhua et al., 2013).

An in-depth analysis of the oxygen vertical profile dynamics is therefore preliminary. The main message we would like to convey here is the methodological approach that can be used to keep monitored the NRT forecast w.r.t. to a benchmark provided by a past simulation.

Regarding nitrate, the scatter plot at 60-100 m erroneously repeated the one for chlorophyll: we report here the corrected version (Fig. R5), where we observe that the operational forecasts are generally in line with the seasonal benchmark (i.e. most of the numbers are within the orange points cloud).

[Figure]

[Figure]

Fig. R5 – Corrected version of Fig. 13: scatter plots of reference (y-axis) versus model forecast (x-axis) for chlorophyll (left column), nitrate (middle column) and oxygen (right column) at different vertical layers: 10-30 m, 60-100 m and 100-150 m. Model forecast are labelled with numbers from 1 to 4 corresponding to lead time from T1 to T4. As benchmark reference, the 2016-2017 pre-operational results are shown for the period of investigation (May to August, orange dots) and for the other periods (yellow dots).

*18. Table 6: I am wondering why the chlorophyll RMSD for the pre-operational is twice as large as the one for the T0 forecast. Do you have an explanation for the nitrate RMSD decrease from T0 to T3?*

REPLY 18 – The limited observations available during the operational period (less than 5 per week) may hinder the statistical significance. In particular for chlorophyll, the qualification period spans 2 years, while the operational one here considered ranges from April to October, when the model errors are much lower (see Fig. 4).

For what concern nitrate, the decrease of its RMSD is related to the limited amount of data available (BGC-Argo floats data may exhibit wide oscillations over subsequent profiles, as shown in Fig. 10).

We will add a comment on this issue, stressing also that robust statistics require much longer time series of data and a larger number of BGC-Argo floats, which is becoming an urgent request for the observing systems to be used in operational biogeochemical oceanography (for both validation and assimilation purposes).

However, the important aspect here is that the quality of biogeochemical variables in the first 4 forecast days stays within a range of 25%, which allows to conclude that the quality of biogeochemical forecast does not degrade and remains satisfactory (i.e., in line with the benchmark) during the first week. This will be better commented at P13-L25 by detailing that "No significant differences can be recognized from the distribution of the four forecast days, showing that the quality of biogeochemical forecast does not degrade during the first week".

Minor/technical comments will be also thoroughly addressed in the review.
Best Regards

**References**

Bittig, H. C., Körtzinger, A., Neill, C., van Ooijen, E., Plant, J. N., Hahn, J., Johnson, K. S., Yang, B., Emerson, S. R. (2018). Oxygen Optode Sensors: Principle, Characterization, Calibration, and Application in the Ocean. Frontiers in Marine Science 4: | DOI: 10.3389/fmars.2017.00429

Clementi, E., Grandi, A., Di Pietro, P., Pistoia, J., Delrosso, D., Mattia, G. (2018). Quality Information Document for MEDSEA_ANALYSIS_FORECAST_PHY_006_013, Copernicus Marine Environment Monitoring Service, available at http://cmems-resources.cls.fr/documents/QUID/CMEMS-MED-QUID-006-013.pdf

Cossarini, G., Salon, S., Bolzon, G., Teruzzi, A., Lazzari, P., and Feudale, L. (2018). Quality Information Document for MEDSEA_ANALYSIS_FORECAST_BIO_006_014, Copernicus Monitoring Environment Marine Service, available at: http://cmems-resources.cls.fr/documents/QUID/CMEMS-MED-QUID-006-014.pdf.

Guyennon, A., Baklouti, M., Diaz, F., Palmieri, J., Beuvier, J., Lebaupin-Brossier, C., Arsouze, T., Béranger, K., Dutay, J.-C., and Moutin, T. (2015). New insights into the organic carbon export in the Mediterranean Sea from 3-D modeling, Biogeosciences, 12, 7025-7046, https://doi.org/10.5194/bg-12-7025-2015.

Macias, D., Stips, A., and Garcia-Gorriz, E. (2014). The relevance of deep chlorophyll maximum in the open Mediterranean Sea evaluated through 3d hydrodynamic-biogeochemical coupled simulations. Ecological Modelling, 281:26-37.

Mattern, J.P., Edwards, C.A., Moore, A.M. (2018). Improving variational data assimilation through background and observation error adjustments. Mon. Weather Rev. 146 (2), 485–501

Mattia, G., Zavatarelli, M., Vichi, M., and Oddo, P. (2013). The eastern Mediterranean Sea biogeochemical dynamics in the 1990s: A numerical study. Journal of Geophysical Research: Oceans, 118(4):2231-2248.

Petihakis, G., Perivoliotis, L., Korres, G., Ballas, D., Frangoulis, C., Pagonis, P., Ntoumas, M., Pettas, M., Chalkiopoulos, A., Sotiropoulou, M., Bekiari, M., Kalampokis, A., Ravdas, M., Bourma, E., Christodoulaki, S., Zacharioudaki, A., Kassis, D., Potiris, E., Triantafyllou, G., Tsiaras, K., Krasakopoulou, E., Velanas, S., and Zisis, N.: An integrated open-coastal biogeochemistry, ecosystem and biodiversity observatory of the eastern Mediterranean – the Cretan Sea component of the POSEIDON system, Ocean Sci., 14, 1223-1245. https://doi.org/10.5194/os-14-1223-2018, 2018.

Richon, C., Dutay, J.-C., Dulac, F., Wang, R., Balkanski, Y., Nabat, P., Aumont, O., Desboeufs, K., Laurent, B., Guieu, C., et al. (2017). Modeling the impacts of atmospheric deposition of nitrogen and desert dust-derived phosphorus on nutrients and biological budgets of the mediterranean sea. Prog. Ocean. 163, 21 - 39. https://doi.org/10.1016/j.pocean.2017.04.009

Tanhua, T., Hainbucher, D., Schroeder, K., Cardin, V., Álvarez, M., and Civitarese, G. (2013). The Mediterranean Sea system: a review and an introduction to the special issue, Ocean Sci., 9, 789-803, https://doi.org/10.5194/os-9-789-2013.

Thierry V., Bittig H., The Argo-Bgc Team (2018). Argo quality control manual for dissolved oxygen concentration. https://doi.org/10.13155/46542

Tsiaras, K. P., Hoteit, I., Kalaroni, S., Petihakis, G., and Triantafyllou, G. (2017). A hybrid ensemble-OI Kalman filter for efficient data assimilation into a 3-D biogeochemical model of the Mediterranean. Ocean Dynamics, 67(6):673-690.

---

## Author Response (AR1)

Dear Editor,
please find here the revision of our manuscript with point-to-point responses to the comments made by the two Reviewers. We wish to thank the Reviewers for the insightful comments that have contributed to improve our manuscript.

In particular:

- The discussion on how the results extracted from metrics application are related with the possible causes of uncertainty (e.g. boundary conditions, physical forcing, accuracy of the reference datasets) has been revised and extended (R2.M2, R1.2, R1.3 and R1.4). The title has been changed from "Marine Ecosystem forecasts: skill performance of the CMEMS Mediterranean Sea model system" to "Novel metrics based on Biogeochemical Argo data to improve the model uncertainty evaluation of the CMEMS Mediterranean marine ecosystem forecasts" (R1.5).
- The definition of the novel metrics used for the model - BGC-Argo comparison have been better presented and discussed (R1.1).
- References to other biogeochemical modelling applications in Mediterranean Sea, and their specific validation framework, have been added (R2.M1).
- Quality of Figures 9, 10 and 12 has been improved and, when feasible, the figures concerning the NRT validation have been updated with the most recent data (R2.9-12 and R2.15).
- Abstract has been modified highlighting some results which were not mentioned in the submitted version (quality improvement along the past decade, the importance of data availability to build robust statistics, the NRT vs seasonal benchmark comparison as a monitoring tool of the operational system).
- Bibliography references have been reviewed.
- Relevant language editing has been performed, wording inconsistencies (as "data set" / "dataset") have been solved, and typos have been corrected (several comments by R1 and R2).

Further, we carefully considered all the other technical points raised by the reviewers.

The points raised by the Reviewers are in *BLACK ITALIC*, our responses in BLUE and the proposed modifications to the manuscript in *GREEN ITALIC*, with line numbers of the revised version. To easily refer through the different reviews, we added a reference to each point raised by the two Reviewers, labelled as "Rx.y", where "x" is the n. of Reviewer, and "y" is a number when referred to a specific comment, or a letter when referred to a technical correction (for the 2 general comments of Reviewer #2, we use R2.M1 and R2.M2). For sake of clarity, when in our replies we refer to specific sentences in the revised manuscript, we use "(RPn,Lm-k)", where "n" is n. of page and "m-k" are specific lines.

Attached to this letter, after the point-by-point responses, we included the revised version of the manuscript with track changes.

**Reply to comments from the Reviewers**

Reviewer #1 – Specific comments

*R1.1. (P8 L1-10) Authors discuss further in the manuscript that direct comparison of model results with the sensor data should be done with caution as the applied corrections may not reflect true value. Thus, application of the introduced metrics which is consistent within itself (e.g. normalising*

*the data to its own surface value) is a good approach. As these are new metrics, for their applicability to other regions, the choice of the criteria deserves a discussion. Do 10% of the surface chl value, or 2mmol/m3 nitrate have a significant meaning? Are these values applicable enough throughout the Mediterranean or have the authors seen regional inconsistencies? Are the strict choice of seasons (jan-mar and apr-oct) valid in practice (can't tell much from Figure 9, but wouldn't using MLDs from the ARGOs yield better estimation of which criteria to use? ARGO profiles show deep chl formation in late Jan and late March 2016 hence much deeper mwb than then model)*

REPLY - The metrics concerning chlorophyll and the relative two periods for the computation are based on published phenomenological understanding. In particular, they have been derived from the outcomes of Lavigne et al. (2015), who identified some standard shapes for chlorophyll profiles from the analysis of a large number of fluorescence data in the Mediterranean Sea (see their Fig. 2). In particular, our summer period is based on the consideration that the DCM profile shape (useful to define our DCM depth index) is typically observed from April to October (Fig. 5 of Lavigne et al., 2015). Then, three other profile shapes (i.e., "homogeneous", "high surface chlorophyll-HSC" and "complex" in Fig. 2 of Lavigne et al., 2015) are characterized by steady-decreasing values with depth, and typically occur between January and March in different Mediterranean regions (Fig. 5 of Lavigne et al., 2015).

The choice of the 10% criterion for the MWB depth index was made after a sensitivity analysis varying the limit between 1 to 10% (not shown). The value of 10% gave results qualitatively consistent with those reported by Lavigne et al. (2015). Further, lower percentage values gave more unclear patterns because the depth of the MWB increases substantially and the thickness of the model layers has also an impact, due to the vertical resolution coarsening (i.e. thickness of model layers is 5 m at 70 m depth, 7 m at 100 m and 10 m at 140 m).

Alternatively, a statistical analysis and additional data (such as Mixed Layer Depth - MLD - values, as suggested by the Reviewer) can provide a better tool to identify the profile shapes and their distribution in time. In particular, the use of the MLD for the choice of seasons' limits surely represents a promising alternative option, since it would account for the specific physical conditions at each float profile, and beside that, it might be useful to catch possible errors propagated by the physical forcing to biogeochemical processes (e.g., comparing model-derived MLD with float-derived MLD). However, it must be noted that there are several possible definitions of MLD which can give slightly different results, thus a sensitivity analysis of the biogeochemical metrics with respect to the MLD definition would be necessary to tune the choice of the criteria based on MLD (this approach is under investigation, and will be developed within the CMEMS R&D activities). The implementation of such methodology might require a considerable amount of data and is out the scope the present work. Therefore, we decided to adopt an a priori, while rigid, temporal subdivision based on literature to test the feasibility of the computation of the novel chlorophyll metrics.

Regarding the criteria for nutricline metrics, they have been designed to track the time evolution of the nitrate vertical profile. Being aware that selecting an unique criterion to detect the nitracline might be sensitive and controversial, we proposed two different approaches: the depth of the 2 mmol/m$^3$ concentration isopleth (NITRCL1) and the depth of the maximum nitrate vertical gradient (NITRCL2). Following Manca et al. (2004), who showed that the values of nitrate concentration at depth higher than 400 m are around 4-5 in the eastern basin and 6-7 mmol/m$^3$ in the western, we considered the isopleth of 2 mmol/m$^3$ a safe value to detect the rapid change between the very low concentration typically measured at the surface and the high concentration

at depth in all areas of the Mediterranean. As discussed for Fig. 10 (see R2.11), the two indexes were able to show different aspects of the nitrate profile evolution, justifying their use to provide indications aimed to monitor the model error behaviour.

We added more details concerning the MWB, DCM and NITRCL1/2 metrics at the end of Section 4.1 (RP8,L27-32 and RP9,L1-9):

*The definitions of DCM and MWB metrics are consistent with the outcomes of Lavigne et al. (2015), who identified some standard shapes for chlorophyll vertical profiles and their temporal distribution from the analysis of a large dataset of fluorescence data in the Mediterranean Sea (see their Figs. 2 and 5). In particular, the summer period defined to estimate the DCM index is based on the consideration that the DCM profile shape is typically observed from April to October. Otherwise, the choice to limit the estimate of the MWB index from January to March is motivated by the fact that steady depth-decreasing profiles typically occur during that period in different Mediterranean regions. Further, the choice of the 10% criterion for the MWB index was set after a sensitivity analysis varying the threshold between 1 to 10% (not shown), with the 10% value giving results qualitatively consistent with those reported by Lavigne et al. (2015).*
*The rationale behind the nitracline depth metrics is defining an index useful to track the time evolution of the nitrate profile. Being aware that the choice of a specific value of nitrate concentration may be controversial, we propose two different indexes: the first is based on the depth of the 2 mmol/m³ concentration isopleth (NITRCL1), the second is related to the depth of the maximum nitrate vertical gradient (NITRCL2). According to Manca et al. (2004), the values of nitrate concentration at depth higher than 400 m are around 4-5 mmol/m³ in the eastern basin and 6-7 mmol/m³ in the western, therefore the 2 mmol/m³ isopleth can be considered a consistent threshold to detect the rapid change between the very low concentration typically measured at the surface and the high concentration at depth in all areas of the Mediterranean Sea.*

*R1.2. (P11 L8-10) Does this partly explain the lower modelled NO3 concentrations (e.g. nwm) due to the lack of N river load time-series? How does model perform in terms of N/P ratios? Does it represent the high N/P ratio character of the Med Sea and its regional differences?*

REPLY - We agree: the effect of the lack of high frequency data of nutrient discharges is one of the most important sources of uncertainty at the daily/weekly time scale (not at seasonal/annual scale) and at very local coastal scales, as discussed by Teruzzi et al. (2018). Indeed, we highlighted this potential issue in the submitted manuscript, and reported in the revised one at (RP12,L13-15). Uncertainty in nwm is partly related to a possible underestimation of the river input forcing and possibly to the effect of lateral circulation from alb and swm1 surface waters (see Fig. 5). A sensitivity analysis of the impacts of the different factors would help in elucidating the most relevant factor. We added a sentence highlighting the specific case of nwm in Section 5.1.1, at (RP10,L13-15):

*Uncertainty in nwm upper layer nitrate (Fig. 5) is partly related to a possible underestimation of the Ebro/Aude/Rhone rivers input forcing and possibly to the effect of lateral circulation from Alboran Sea and Southern Western Mediterranean surface waters (see Fig. 5, panels "alb" and "swm2").*

An important aspect to consider when discussing the model performance is that the fine subdivision of the Mediterranean Sea in 16 sub-basins allows to detect the relevant spatial gradients and, thus, to highlight possible issues, as discussed in the submitted version, and reported in the revised one at (RP17,L28-32 and RP18,L1-3). The choice of the 16 sub-basins, designed by the Med-BIO component of Med-MFC and agreed as common skill assessment protocol for the other two components (Med-PHY and Med-WAV), was a trade-off among having a number of areas as larger as possible, the need for having robust in situ statistics and the known characteristics of the dynamics derived from literature. Thus, our proposed subdivision is a relevant result in itself. We included this point in the Discussion on the sub-basin subdivision at (RP18,L3-5), adding a comment on the detection of the surface nitrate underestimation of Fig. 5:

*[…], justifying a posteriori our sensible definition of the 16 sub-basins. In fact, the comparison of nutrients profiles of Fig. 5 highlights the satisfactory model performance in reproducing the mean spatial gradients and the possible anomalies, such as the underestimation of upper layer nitrate in nwm sub-basin.*

Concerning N/P ratios, the performance of OGSTM-BFM model system was assessed in Lazzari et al. (2016) for nitrate and phosphate separately, showing a general higher-than-Redfield ratio in the Mediterranean Sea (closer to a N:P=22 ratio, with exception of the Alboran Sea area, characterized by a lower-than-Redfield ratio) and a significant spatial variability. The present operational configuration of the MedBFM incorporates the results presented by Lazzari et al. (2016; see their Fig. 7 here reported as Fig.R1, for further details please refer to the paper) and provides consistent results on N:P ratio (see also N and P values in Fig. 5).

[Figure]

Fig.R1 - Results from Lazzari et al. (2016), model-derived vertical profiles of phosphate (x-axis) and nitrate (y-axis) averaged by sub-basins (reported in Fig. 1 of Lazzari et al., 2016) for the period 1999–2004. Each entry represents the spatial-temporal average at a certain depth: 0–50 m layer (circles) and 50-bottom layer (triangles).

*R1.3. (Fig9) How does BGC-Argo surface chl compare with the satellites? P11 L4 suggest the model has a higher (0.015) bias for the winter (model vs satellite), supported by Fig8 with more pronounced bias for the west/northwest Med, while paragraph of P11 L32 suggest the model has lower values when compared to BGC-Argo data. Is there a consistent ratio between satellite and float data, and how applicable is it to use global correction of division by 2 as suggested by Roesler et al. (2017) taking into account the regionally different ratios shown in the same article. As the Mignot et al. (2018) manuscript is in review, I cannot comment about their results but can the application of their method suggest different correction factors with a better regional fit?*

REPLY – We thank the Reviewer for having raised this point, since the inconsistency in surface chlorophyll observations between satellite and BGC-Argo floats has been already observed in our investigations (see Fig.R2, showing that positive deviations of BGC-Argo floats w.r.t. satellite data are evident in winter, while negative deviations of BGC-Argo data are present in summer and spring seasons) and represents a potential issue, not only for validation purposes as discussed here, but also for multi-platform data assimilation (e.g., combined assimilation of chlorophyll from BGC-Argo floats and satellite). Regional corrections of BGC-Argo floats data can be advisable (as shown in Roesler et al., 2017), such as regional algorithms for the ocean color exist for satellite, to provide chlorophyll in different regions of the global ocean. However, at the present stage, as shown by Roesler et al. (2017), regional calibration factors appear to be based on very limited statistics: for this reason, we think it is more robust to rely on the recommend factor of 2 global bias correction.

Methods such as that of Mignot et al. (2019) can be helpful to give a better estimate of the observational error (of great importance in data assimilation), proven the availability of a sufficient amount of in situ independent data.

Such investigation is off-topic with respect to our manuscript, but emphasizes that operational systems are optimal tools to test the consistency of many different sources of information, as was already reported in the submitted manuscript citing She et al. (2016), and in the revised version at (RP2,L25-26) and (RP18,L25-26).

[Figure]

Fig. R2 - Chlorophyll concentration comparison between surface BGC-Argo floats data and satellite data (CMEMS ESA-CCI Mediterranean data). Data of all floats available between 2015-2017 are here included. The matchup between data is performed on a daily basis and using a bilinear spatial interpolation.

*R1.4. (P14 L20-23) Authors point out that the introduced BGC-ARGO related metrics are already being implemented for data assimilation purposes with consequent improvements in model*

*solutions. Before the assimilation phase (e.g. pre-opetaional runs), does the skill assessment documented here (of BGC-Argo metrics) reveal any prior messages for model parameter adjustment such as for light attenuation or nutrient assimilation rates, or errors of physical model origin? I can see the use of this dataset not only for forecasting purposes and skill assessment purposes, but its high resolution coverage including ocean interior is of high value. A short comment on that would be good scientific addition to the manuscript.*

REPLY – We thank the Reviewer for this comment. The integration of BGC-Argo floats within an analysis and forecasting system (in terms of data assimilation) paved the way to an in-deep study of the interior of the sea and its dynamics. Then, given the fact that the BGC-Argo also provide profiles of PAR, salinity and temperature, which are forcing mechanisms of the biogeochemical processes (or proxies for the forcing impacting the biogeochemistry), an analysis of the model uncertainty can be made using multivariate statistical analysis, with the aim of disentangling the sources of error on biogeochemical profiles. This point has been also reported in comment R2.M2 (e.g., issues related to inaccuracy of the boundary conditions, oxygen float data quality, model surface chlorophyll discrepancy).
Beside the validation, it is worth to note that specific physical (such as MLD and euphotic zone depth) and biogeochemical (such as NITRCL1/2, DCM and MWB) indexes that can put in relation the shape and intensity of the profiles with the underlying processes would allow to better investigate coupled vertical physical-biogeochemical processes. We think that our work provides a first step to identify and quantify several functional biogeochemical indexes. However, a critical point is the availability of a sufficient amount of profiles for variables like nitrate and oxygen, which may allow for statistically significant analysis.
As also discussed in R2.M2 reply, we included a short comment on this point in the Discussion section (RP18,L26-34):

*In perspective, the integration of BGC-Argo within operational ocean forecasting systems in terms of data assimilation (see Cossarini et al., 2019) becomes strategical for an in-deep study of the interior of the sea and its dynamics. Moreover, considering that BGC-Argo floats also provide profiles of physical quantities (i.e., radiometric quantities – PAR- and temperature), an analysis of specific physical (e.g., MLD, euphotic zone depth) and biogeochemical (e.g., NITRCL, MWB, DCM) indexes that can reveal relationships between the shape and/or intensity of the profiles and the underlying dynamics would allow to further delve into coupled vertical physical-biogeochemical processes. In such a view, our work provides a first step to identify and quantify several functional biogeochemical indexes. Nonetheless, a critical point remains the availability of a sufficient amount of profiles for variables like nitrate and oxygen, which may allow for statistically significant analysis.*

*R1.5. I see that the manuscript is designed as a document for the overall skill assessment of MedBFM, but both the abstract and the manuscript throughout have stressed the importance and usefulness of their new metrics (GODAE Class 1 and 4, and especially the use of BGC-Argo), and I agree with them, and these sections of the manuscript stand out as the novel scientific content. The title fails to give this message and won't promote this novel scientific content of the manuscript. I leave it to the authors consideration.*

REPLY – We agree: we changed the title highlighting the novel metrics and the usefulness of BGC-Argo float data. The title now is: "Novel metrics based on Biogeochemical Argo data to improve the model uncertainty evaluation of the CMEMS Mediterranean marine ecosystem forecasts".

Reviewer #1 – Technical corrections

R1.a)    (P2 L1) As a user-driven
         Corrected adding "a".

R1.b)    (P3 L21) semi-labile
         Corrected adding "-".

R1.c)    "In situ" appear as in-situ in various parts of the text. Ocean science journal asks for the use "in situ".
         Thanks: the revised text includes now only "in situ".

R1.d)    (P7 L16) replace relays by relies
         Corrected.

R1.e)    (P7 L20) remove first "run"
         Removed.

R1.f)    (P8 L8) Add CORR acronym
         The "CORR" acronym has been added.

R1.g)    (P9 L24) model simulates well
         Corrected.

R1.h)    (P18 L1) represent a useful
         Corrected.

R1.i)    (Fig10) Color code for NITR1 and NITR2 is missing
         Thanks: we added the color for NITRCL1 (blue) and NITRCL2 (red). See also point R2.11.

R1.j)    (Fig13) Authors used CHL for 60-100 m twice, suggesting NO3 is missing at that depth range in the figure.
         Thanks: the scatter plot for nitrate at 60-100 m erroneously repeated the one for chlorophyll. Fig.13 has been corrected in the revised version (see also point R2.17).

Reviewer #2 – General comments

*R2.M1 It doesn't give an overview of other published simulations on the Mediterranean Sea. The reader is not able to appreciate the quality of this simulation in relation to other BGC models in the literature;*

REPLY – We agree: the revised version refers to other biogeochemical model applications in Mediterranean Sea. In particular, considered the focus of the manuscript (i.e., biogeochemical operational forecast and its uncertainty estimates), we have included the Poseidon operational system (HYBRID-POM-ERSEM model, Tsiaras et al., 2017; Petihakis et al., 2018) at (RP2,L31-33):

*This is applicable to the Mediterranean Sea operational systems which include, besides CMEMS Med-MFC, also the Poseidon operational system built on the HYBRID-POM-ERSEM model coupling (Tsiaras et al., 2017; Petihakis et al., 2018).*

Besides that, there are other recent publications describing multi-annual simulations (a not exhaustive list includes Mattia et al., 2013; Macias et al., 2014; Guyennon et al., 2015; Richon et al., 2017) but we think it is out of the scope of this manuscript to list all the models and to provide an overview of their characteristics.

Nevertheless, given that the focus of our manuscript is on validation, in the revised version we have added a synthetic overview of the data availability for validating biogeochemical simulations in the Mediterranean Sea. The overview shows that the most used variable for validation is satellite-derived surface chlorophyll (Tsiaras et al., 2017; Mattia et al., 2013; Macias et al., 2014; Guyennon et al., 2015; Richon et al., 2017). Further, in situ measurements from vessels and scientific cruises are also used in Richon et al. (2017) and Guyennon et al. (2015), but allow only to validate very limited time and space portions of the simulations. Only a few basin-wide validation frameworks, especially for nutrients, are based on comparison with climatology (e.g. Tsiaras et al., 2017, used a seasonally aggregated reference for the whole Mediterranean Sea built on 1990-1999 data from SeaDataNet). Very rarely the vertical proprieties of biogeochemistry are assessed (e.g. Guyennon et al., 2015 and Teruzzi et al., 2014).

The novelty of our work lies in the introduction of the use of BGC-Argo floats data as observational counterpart for the model validation, and presents a comprehensive validation framework from basin-wide and seasonal scale to mesoscale and weekly scale (following GODAE recommendations), with an accuracy level depending on the specific variable and the availability of reference data.

The following paragraph has been added in the Introduction (RP3,L1-14):

*More in general and concerning biogeochemical applications in the Mediterranean Sea, the limited availability of observational reference data often hinders the validation assessment of model products. The most common approach is based on contrasting model outputs with satellite-derived surface chlorophyll (Tsiaras et al., 2017, for year 2000; Mattia et al., 2013; Macias et al., 2014; Guyennon et al., 2015; Richon et al., 2017, for a portion or the whole respective investigated multi-year periods). In situ measurements from vessels and scientific cruises are also used in Richon et al. (2017) and Guyennon et al. (2015), but allow only to validate very limited temporal and spatial subsets of the simulations (i.e., time series of fixed stations or single transects in a very confined time range). On the other hand, a few basin-wide validation frameworks, especially for nutrients, are based on comparison with climatology, e.g. Tsiaras et al. (2017) used a seasonally aggregated reference for the whole Mediterranean Sea built on 1990-1999 data from SeaDataNet. Generally, modelled vertical proprieties of biogeochemistry are rarely assessed (e.g. Guyennon et al., 2015 and Teruzzi et al., 2014) due to the lack of adequate reference datasets. In the recent years, the availability of biogeochemical vertical profiles in the Mediterranean Sea has significantly increased with the deployment of Biogeochemical Argo floats (hereafter BGC-Argo floats; Johnson and Claustre, 2016), whose datasets constitute an unprecedented source of reference for*

*biogeochemical model skill assessment, spanning from basin-wide and seasonal scale to mesoscale and weekly scale.*

*R2.M2 It doesn't manage to rationalize the discrepancies diagnosed by the comparison with observations and to disentangle the origin of the observed biases (problems due to the physical model, the biogeochemical parameterizations or to the sensor quality control?). They are a lot of metrics that are not enough used to suggest some solutions and to improve the future systems*

REPLY – We thank the Reviewer for this comment. The submitted manuscript discussed a number of general weaknesses of the MedBFM model system (also reported in the revised version): (RP12, L3-5) refer to the high winter chlorophyll RMSD values in the nwm sub-basin, relating it with possible drawbacks in the reproduction of vertical mixing included in the sub-grid parameterization of Med-PHY component; (RP14, L5-10) refer to discrepancies in oxygen estimation, which may be related with possible inconsistency of the QC procedure of BGC-Argo oxygen data (see also R2.12); (RP19, L1-15) refer to the decreased skill at the domain boundaries and at the coastal areas. Some of these will be objects of our future development work, as already planned within Med-MFC (e.g. vertical mixing scheme in Med-PHY, oxygen formulation in Med-BIO), and also requested by the CMEMS "continuous improvement" paradigm.

In the revised version we have enriched the comments related to the results gathered from the metrics application and coherently related them with our analysis on possible causes. In particular, we have improved our discussion by reporting and rationalizing the following cases where our validation framework can detect specific discrepancies:

a) As reported at point R1.2, we highlight that our definition of the 16 sub-basins revealed to be a sensible choice, since it allowed to demonstrate the good model performance in reproducing the mean spatial gradients within the domain (Fig. 5), and the possible anomalies (as the one in nwm, see following item). We include the point related to the domain subdivision in the Discussion at (RP18,L3-5), adding a comment on the detection of the surface nitrate underestimation of Fig. 5: *[…], justifying a posteriori our sensible definition of the 16 sub-basins. In fact, the comparison of nutrients profiles of Fig. 5 highlights the satisfactory model performance in reproducing the mean spatial gradients and the possible anomalies, such as the underestimation of upper layer nitrate in nwm sub-basin.*

b) Again as in R1.2, we relate the underestimation of nitrate in the nwm sub-basin in the subsurface layer partly to possible underestimation of the terrestrial input and partly to the impact of the incoming Atlantic waters. This point is included in Section 5.1.1, at (RP10,L13-15): *Uncertainty in nwm upper layer nitrate (Fig. 5) is partly related to a possible underestimation of the Ebro/Aude/Rhone rivers input forcing and possibly to the effect of lateral circulation from Alboran Sea and Southern Western Mediterranean surface waters (see Fig. 5, panels "alb" and "swm2").*

c) Another detected discrepancy that we have included in the revised version is the presence of some overestimation for the oxygen field between model and BGC-Argo profiles below 200 m (as shown in Tab.5 and in the revised Fig. 11, with an extended value range which better illustrates the vertical gradients, see also point R2.12). Since the quality check protocol of oxygen data from BGC-Argo floats is continuously progressing, we envisage the need for a specific study regarding the oxygen validation and the analysis of oxygen variability simulated by the MedBFM model. A sentence has been added in Section 5.1 at

(RP18,L19-20): *A specific investigation focused on the oxygen validation framework and the analysis of the oxygen variability simulated by the MedBFM model is in preparation.*

d) As also reported at point R1.4, the metrics based on BGC-Argo data (e.g., DCM and MWB) are very innovative and informative. BGC-Argo floats provide simultaneous measurements of physical variables, such as profiles of PAR and temperature, that act as forcing of the biogeochemistry processes (or proxies for the forcing impacting the biogeochemistry). Thus, an integrated analysis of the model-observation uncertainty can be made comprehensively using multivariate analysis (e.g. PCA and neural network methods), with the aim to disentangle the error sources on vertical profiles. Moreover, we have to note that such analysis would need not just simple measures of distance between observations and model values (such as BIAS and RMSD), but indexes that can put in relation the shape and intensity of the biogeochemical profiles with the underlying processes (light limitation, temperature kinetic dependencies, nutrient availability through vertical fluxes, vertical dispersion through mixing). Within this framework, we think that our work (and the proposed metrics) provides a first step towards the identification and quantification of several functional indexes. Beside this consideration, another critical point is the availability of a sufficient amount of float profiles for variables like nitrate and oxygen, which would be required for a statistically significant analysis covering most of the Mediterranean regions and a longest period of the year. The proposed comment is at point R1.4, included in the Discussion at (RP18,L26-34), and here reported for sake of clarity: *In perspective, the integration of BGC-Argo within operational ocean forecasting systems in terms of data assimilation (see Cossarini et al., 2019) becomes strategical for an in-deep study of the interior of the sea and its dynamics. Moreover, considering that BGC-Argo floats also provide profiles of physical quantities (i.e., radiometric quantities – PAR – and temperature), an analysis of specific physical (e.g., MLD, euphotic zone depth) and biogeochemical (e.g., NITRCL, MWB, DCM) indexes that can reveal relationships between the shape and/or intensity of the profiles and the underlying dynamics would allow to further delve into coupled vertical physical-biogeochemical processes. In such a view, our work provides a first step to identify and quantify several functional indexes. Nonetheless, a critical point remains the availability of a sufficient amount of profiles for variables like nitrate and oxygen, which may allow for statistically significant analysis.*

e) Regarding the chlorophyll satellite-model comparison we extended the sentence in Section 5.1.1 (RP10, L3-8), adding more details on the discrepancies of the bloom model appearance in the north-western Mediterranean, with specific references, and coherently moving a sentence concerning the same issue from Section 5.1.2. Here we hypothesize a possible link with the physical dynamics, however the specific investigation is left out of the present contribution.
*Finally, modelled late winter-early spring surface chlorophyll maxima in nwm appear anticipated of 2-3 weeks w.r.t. satellite ones: this is related to a possible mismatch of the spatial patterns which characterize the temporal succession of deep convection and subsequent stratification and bloom, known to have a very high patchy (i.e., at mesoscale and sub-mesoscale) dynamics in this area (Estrada et al., 2014; Mayot et al., 2017; Severin et al., 2017). The magnitude, timing and spatial pattern of such mesoscale and sub-mesoscale structures might not be completely well resolved, thus resulting in increased discrepancies with observations.*

Reviewer #2 – Specific comments

*R2.1. P3 L26-27: "surface chlorophyll . . . of CMEMS" Which CMEMS product? Is it the same as the one used for assessment on Fig 3 and Fig 4?*

REPLY – The product is the OCEANCOLOUR_MED_CHL_L3_NRT_OBSERVATIONS_009_040 as described in section 2.2 ("Set up of the pre-operational qualification simulation for Med-BIO") and section 3 ("Reference datasets for validation").
Moreover, it is important to remind (as explained in the submitted version, and reported in the revised one at RP8,L6-8 and RP11,L29-31) that this product is used in the data assimilation scheme (at the beginning of each weekly run at day T-7) and then to assess the skill of the forecasts (as in Fig. 3 and 4 for the pre-operational run, and in Fig. 12 for the forecast run) before it is used for assimilation at the next weekly run. Following Mattern et al. (2018), the root mean square (RMS) of the differences between observation and prior model solution provides a measurement of the data assimilation performance and represents a short-term forecast skill metric since it is based on observations that are going to be assimilated in the following cycles. We think that this concept is already clearly depicted in the text. However, we are open to add details if the Reviewer suggests they are needed.

*R2.2. P4 L10: Which database do you use for river inputs? Reference?*

REPLY – Details on the boundary conditions for river nutrients inputs were reported in Section 2.2, 5th and 6th items, and reported in the revised version at (RP6,L3-11), and derive from the dataset built during the PERSEUS FP7 project. In the revised version, we include that the dataset is based on a total of 39 rivers with runoff larger than 50 $m^3$/s (RP6,L3). To the best of our knowledge, this deliverable represents the most up-to-date information about terrestrial nutrient discharges for the Mediterranean basin.
To further improve readability, in the revised version we have added the reference to the river details described in Section 2.2 also in Section 2.1 at (RP4,L28).

*R2.3. P4 L15-16: "Additional 2D fields include the surface data for solar irradiance and wind stress": Where are these data from?*

REPLY – These data are provided via the offline coupling with MED-PHY component, which derives the atmospheric forcing from the 6-hours ECMWF operational analysis and forecast fields at 0.125° horizontal-resolution. For details please refer to CMEMS catalogue[1]. Clarifications about the source of the 2D fields have been added to the revised Section 2.1 at (RP5,L1-3):

*Additional 2D fields from MED-PHY include the surface data for solar shortwave irradiance and wind stress (derived by the ECMWF atmospheric forcing, see details later), which are used, respectively, as input for the BFM optical module and to solve the gas air-sea exchanges.*

*R2.4. P5 L11: Is the spin-up forced by a climatological year?*
* * *
[1] http://marine.copernicus.eu/services-portfolio/access-to-products/?option=com_csw&view=details&product_id=MEDSEA_ANALYSIS_FORECAST_PHY_006_013

REPLY – The spin-up is based on year 2016 run without data assimilation. This procedure is coherent with that adopted for the spin-up of the pre-operational run of the physical component of the Med-MFC (see details in Clementi et al., 2018). We think it is not necessary to add further details to the revised text (RP5,L26-27), but we are open to do it in case the Reviewer will find it useful.

*R2.5. P6 L16-20: is it the same dataset as the one used in the data assimilation scheme? Can you precise the temporal frequency. Daily?*

REPLY – Yes, the dataset is the same, and it is used at daily frequency. We have added "daily" in Section 3 at (RP7-L2) to better clarify this point. As discussed at point R2.1, validation is based on satellite surface chlorophyll data that are not yet assimilated.

*R2.6. P6 L24-30: do you use the BGC-Argo dataset available on Coriolis website?*

REPLY – We download our data from LOV database due to specific quality control developed in the frame of the CMEMS Service Evolution project MASSIMILI (see details at https://www.mercator-ocean.fr/en/portfolio/massimili-2/). The use of BGC-Argo floats data from Coriolis and CMEMS INSITU-TAC is presently under consideration.

*R2.7. P11 L19: prefer 'averaged' instead of 'integrated'*

REPLY – We agree with the suggestion, we have substituted 'integrated' with 'averaged' at (RP12,L24).

*R2.8. P11 L20 and Fig9: why do the correlation values vary so abruptly and reach zero sometimes in winter?*

REPLY – We thank the Reviewer for having raised this issue. The plots of Fig. 9 (right panels in the submitted version) were erroneously computed including also dates without float data (see also R2.10). In other few cases, very low correlation values were related to some possible inconsistencies in the float measurements (see for example the isolated chlorophyll values up to 0.20 mg/m$^3$ below 200 m in April 2016 and in May 2017). Our operational check of the reliability of the input data is still progressing. Thus, in these cases, we safely decided to remove from the computation these profiles. A corrected and improved version of Fig. 9 has been included in the revised version.

*R2.9. P11 L22-23 and Fig9: This sentence is not convincing. Why these differences between modelled and observed depth of MWB? What is the difference between the mixed layer depth (MLD) and the depth of MWB? Why did you choose this index? Did you compare the mixed layer depths along the float trajectory?*

REPLY – We thank the Reviewer for this comment. We introduced the MWB in order to identify an index quantifying the thickness of the layer affected by the surface chlorophyll winter bloom. As commented in the Reply to R2.M2 (item "d"), this is a first step towards a specific metric to keep track of biogeochemical processes and their relationship with physical drivers.

As also explained in the reply to R1.1, the definition of MWB is based on the phenomenological results reported by Lavigne et al. (2015), who identified some standard shapes of chlorophyll profiles analyzing a large number of chlorophyll profiles (from fluorescence data) in the Mediterranean Sea (see their Fig. 2). Three of those profile shapes (i.e., "homogeneous", "HSC" and "complex") are characterized by decreasing-with-depth values and are typically observed during the winter months in different Mediterranean regions (Fig. 5 of Lavigne et al., 2015). They are not necessarily limited by the MLD, since other factors (e.g., light and nutrient conditions) play a role in the surface bloom dynamics. In addition, there are several possible definitions of MLD which can give slightly different results, so a sensitivity analysis of the biogeochemical metrics to the MLD definition would be necessary to tune the choice of the criteria based on MLD.

The specific choice of the limit of 10% was made after a sensitivity analysis varying the limit between 1 to 10% (not shown). The value of 10% gave results qualitatively consistent with those reported by Lavigne et al. (2015), whilst lower percentage values gave more unclear patterns because the depth increases substantially and the thickness of model layers has also an impact, due to the vertical resolution coarsening (i.e. thickness of model layers is 5 m at 70 m depth, 7 m at 100 m and 10 m at 140 m). Further, the MWB metric fails to capture the "modified DCM" shape profile (as defined in Lavigne et al., 2015) which occurred for float 6901653 during winter 2016 (Fig. 9). Hence, given the constraint in the definition and the vertical discretization of the model, the application of this index to the floats data may originate some under- or overestimations and uncertainty of a few decameters (see Tab. 3). Despite these limitations, we think that the MWB represents a feasible and informative metric to be coupled with the DCM metrics.

In the revised manuscript, we have better explained the scientific rationale of the MWB metrics in Section 4.1 (RP8,L27-32 and RP9,L1-2), here also reported for sake of clarity:

*The definitions of DCM and MWB metrics are consistent with the outcomes of Lavigne et al. (2015), who identified some standard shapes for chlorophyll vertical profiles and their temporal distribution from the analysis of a large dataset of fluorescence data in the Mediterranean Sea (see their Figs. 2 and 5). In particular, the summer period defined to estimate the DCM index is based on the consideration that the DCM profile shape is typically observed from April to October. Otherwise, the choice to limit the estimate of the MWB index from January to March is motivated by the fact that steady depth-decreasing profiles typically occur during that period in different Mediterranean regions. Further, the choice of the 10% criterion for the MWB index was set after a sensitivity analysis varying the threshold between 1 to 10% (not shown), with the 10% value giving results qualitatively consistent with those reported by Lavigne et al. (2015).*

Moreover, we have modified the original sentence in Section 5.1.2 according to this comment (RP13,L1-6):

*Considering the constraint in the definition of the MWB depth and the vertical discretization of the model, the application of such index to floats data may originate some inconsistency (as shown for winter 2016 in Fig. 9), and under- or overestimations and uncertainty of a few decameters (see Tab. 3). Despite these limitations, we consider the MWB as a feasible and informative metric alongside the DCM metrics to characterize the seasonal chlorophyll profile evolution.*

*R2.10. Fig9: in summer 2017, there is no diagnosed DCM for model. Why?*

REPLY – We thank the Reviewer: there was an error in the plot (see also point R2.8). The revised version of Fig. 9 reports a wider y-axis in the bottom panel, and we observe that the DCM in 2017 appears shallower than in 2016 (please also consider the float trajectory – in the top panel of Fig. 9 – that covers the south-western Mediterranean from Balearic Islands to Alboran Sea, which is known to have a shallower DCM, see Lazzari et al., 2012).

*R2.11. Fig 10: skill index 4th row: the legend is missing for NITRCL1/NITRCL2. Why is the blue index missing for model in the beginning of 2017?*

REPLY - We thank the Reviewer: we have corrected the caption (NITRCL1 is in blue and NITRCL2 is in red). We have also extended the maximum depth to 300 m (new version of Fig. 10), which allows to recognize the agreement between NITRCL1 values computed on model and BGC-Argo data at the beginning of 2017. It can be observed that in the period April-July 2017 one of the two nitracline indexes (i.e., NITRCL2) computed on model output is much shallower than the one estimated from the BGC-Argo float data. Thus, the two indexes were useful to show different aspects of the nitrate profile evolution, justifying their use to provide indications aimed to monitor the model error behaviour.
A comment has been added in the revised version in Section 5.1 at (RP13,L16-19):

*The model NITRCL1 and 2 perform generally good, however, it can be observed that in the period April-July 2017 the NITRCL2 appears much shallower than what estimated by the float data. The two indexes show different aspects of the nitrate profile evolution, justifying their use to provide indications aimed to monitor the model error behaviour.*

*R2.12. Fig10-11: Authors don't show depths below 300m. For oxygen, the colorbar is saturated below 200 umol/L. It doesn't allow to study the nitrate gradient and the OMZs.*

REPLY – The model comparison for depths below 300 m is shown in Fig. 5 for nutrients and oxygen for 6 selected sub-basins. Given the slow dynamics of the deep layers, the comparison based on climatology (GODAE Class 1 metrics with NODC-OGS dataset) demonstrates the capability of the model to reproduce the nutrient values along the Mediterranean Sea gradient and the relative minimum of the oxygen concentration below 400 m.
The depth of the relative minimum of oxygen displayed in Fig. 5 is consistent with published information: according to Tanhua et al. (2013; see their Fig. 6), the oxygen minimum layer (OML) core in Mediterranean is located at 500 - 700 m in the eastern basin, "well below the layer of maximum S [salinity] occupied by the LIW" (see sub-basins ion2, ion3 and lev4 in Fig. 5), whilst in the western basin it is shallower at around 400 m depth (see sub-basin nwm in Fig. 5; other sub-basins can be found in Cossarini et al., 2018). A sentence about such agreement with the historical data has been added in Section 5.1 at (RP10,L27-30):

*We can observe that the depth of the relative minimum of oxygen displayed in Fig. 5 is consistent with the cruise data shown by Tanhua et al. (2013): the oxygen minimum layer core in the eastern basin is located below 500 m (sub-basins ion2, ion3 and lev4 in Fig. 5), whilst is at around 400 m*

*depth in the western basin it (see sub-basin nwm in Fig. 5; for the sub-basins please refer to Cossarini et al., 2018).*

On the other hand, BGC-Argo floats allow to observe relatively fast dynamics (i.e., at weekly time scale according to their current 5-day sampling frequency) which are particularly relevant for the biogeochemical processes in the upper layer. Thus, we used BGC-Argo floats data to verify the model capability to reproduce the mesoscale and vertical dynamics in the upper layer at the weekly time scale. This represents a very stringent comparison, which allowed to demonstrate the timing and intensity of the evolution of signals that track biogeochemical processes in the upper layer. Therefore, we think that it would be not enough illustrative to compare and show deep profiles for model and float, also considering the ongoing and progressing advancements of the quality control process for BGC-Argo floats data. In fact, only very recently a new product quality control procedure (following Bittig et al., 2018 and Thierry et al., 2018) has been started to be implemented for oxygen to correct biases on sensor: to the best of our knowledge, it is not yet available for all floats in the Mediterranean Sea. Thus, we prefer to keep the maximum depth of Figs. 10 and 11 at 300 m to better exploit the model-float visual comparison in the upper layer (roughly identified with the euphotic layer, generally not deeper than 200 m), where most of the biogeochemical processes occur.

We have revised the colorbar of Fig. 11 starting from oxygen value at 180 mmol/m$^3$: the new Figure 11 displays some discrepancies (i.e., a bias of about 10-20 mmol/m$^3$) that were already commented along with Table 5. However, given the aforementioned ongoing quality improvements in the BGC-Argo float oxygen data, a quantitative calculation of the discrepancy between model and floats (as done for the chlorophyll and nitrate metrics) has not been defined yet, and we proposed only qualitative considerations, as reported at (RP14, L5-10).

In the revised manuscript, we have better clarified the limitations of the BGC-Argo data use for oxygen, adding a sentence in Section 5.1 (RP14,L10-12):

*Only very recently a new product quality system (following Bittig et al., 2018 and Thierry et al., 2018) has started to be implemented for oxygen data to correct biases on sensor: to the best of our knowledge, it is not yet available for all floats in the Mediterranean Sea.*

Finally, we would like to point out that with improving oxygen data quality, the use of BGC-Argo floats to investigate quantitatively the simulated oxygen will become very promising for evaluating surface and deeper layer dynamics. A comment on the improvement of the oxygen validation framework has been added in the discussion regarding future developments at (RP18,L19-20), as already proposed at item "c" of R2.M2:

*A specific investigation focused on the oxygen validation framework and the analysis of the oxygen variability simulated by the MedBFM model is in preparation.*

*R2.13. P12 L21: I would say 'solubility' instead of 'saturation'*

REPLY – Thanks, the term "solubility" has been used in the revised version.

*R2.14. P12 L22: What are the consumption terms?*

REPLY – Comsumption terms are defined as respiration terms by bacteria and plankton community (4 phytoplankton and 4 zooplankton groups). We have added the definition in the revised version (RP13,L33-34):

*[…] presence of consumption terms (defined as respiration terms by bacteria and plankton community: 4 phytoplankton and 4 zooplankton groups).*

*R2.15. Fig 12: needs to be improved (quality). Do you compare to the daily NRT L3 chlorophyll product? The figure seems to be at a coarser temporal resolution. Moreover, the yaxis labels don't appear correctly (they are truncated). I don't understand why the red curve is constant during the summer.*

REPLY – We agree about the poor quality of Fig.12: an updated version (using bar plot and extending the time range until January 2019) is present in the revised document. The comparison is made, for each week, between the first 3 days of forecast (black for T1, blue for T2 and green for T3) and the corresponding daily NRT L3 satellite product (as written at RP14,L24). The red line represents the seasonal benchmark defined as the averages of the RMSD shown in Fig. 8. Thus, the 2 seasonal benchmarks give information on the average quality of forecast for the 2 periods.

*R2.16. P13 L15: 0.041 instead of 0.41 mg/m3?*

REPLY – Thanks: yes, it is 0.041 mg/m$^3$. It has been corrected in the revised version (RP14,L29).

*R2.17. P13 L18-20 and Fig 13: Authors don't comment much the Fig13. For example, the nitrate dots are very scattered (between 0 and 100m depth), do you think that it is due to nitrate sensor anomalous values or is it a problem in the physical or biological models? Oxygen panels display a bias (model overestimates float data): do you think it is a bias in the model or in the data?*

REPLY – We acknowledge the Reviewer for this point: the comment to Fig. 13 has been further enriched in the revised version (RP15,L3-12). As we pointed out in the Discussion of the submitted manuscript (reported in the revised one P18,L16-17), and also thoroughly commented at R2.12, the systematic bias for oxygen at depth can be either due to both model and data uncertainty. Only very recently, new product quality procedure (following Bittig et al., 2018 and Thierry et al., 2018) has started to be implemented, but up to now is not yet available for all floats in the Mediterranean.
On the other hand, concentrations of oxygen lower than 180 mmol/m$^3$ in subsurface layer (100-150 m) appear quite suspicious for the Mediterranean Sea (see Manca et al., 2004, and also Tanhua et al., 2013).
An in-depth analysis of the oxygen vertical profile dynamics is therefore preliminary. The main message we would like to convey here is the methodological approach that can be used to keep monitored the NRT forecast w.r.t. to a benchmark provided by a past simulation. We have added in Section 5.2, at (RP15,L3-4) the following sentence, which introduces Fig. 13:

*To provide a monitoring of the quality of the NRT forecast with respect to a seasonal reference defined by the pre-operational qualification run, […]*

And, at (RP15,L10-12):

*We can observe that floats oxygen concentrations in the subsurface layer (100-150 m) are lower than 180 mmol/m³, which appears quite anomalous for the Mediterranean Sea (see Manca et al., 2004, and also Tanhua et al., 2013), thus conveying a suspect instrumental bias of the oxygen sensor, as already discussed in Section 5.1.2 for Fig. 11.*

Regarding nitrate, the scatter plot at 60-100 m erroneously repeated the one for chlorophyll: in the corrected version we observe that the operational forecasts are generally in line with the seasonal benchmark (i.e. most of the numbers are within the orange points cloud).

*R2.18. Table 6: I am wondering why the chlorophyll RMSD for the pre-operational is twice as large as the one for the T0 forecast. Do you have an explanation for the nitrate RMSD decrease from T0 to T3?*

REPLY – We thank the Reviewer for this comment, since it helped to better discuss the results of Tab. 6. The limited observations available during the operational period (less than 5 per week) may hinder the statistical significance. In particular for chlorophyll, the qualification period spans 2 years, while the operational data considered for Tab. 6 ranges from April to October, when the model errors are much lower (see Fig. 4).
For what concern nitrate, the decrease of its RMSD is related to the limited amount of data available (BGC-Argo floats data may exhibit wide oscillations over subsequent profiles, as shown in Fig. 10). The important aspect here is that the quality of biogeochemical variables in the first 4 forecast days ranges within ±25%, which allows to conclude that the quality of biogeochemical forecast does not significantly degrade and remains satisfactory (i.e., in line with the benchmark) during the first week.
Following such considerations: the comment to Tab. 6 has been mostly revised (RP15,L13-18):

*The RMSDs of the four forecast days (Tab. 6) remain within a range of ±25%, generally showing that the quality of biogeochemical forecast does not significantly degrade during the first week. More precisely, chlorophyll and oxygen RMSD of T3 and T4 are slightly larger than T1, while nitrate RMSD of the last forecast days is lower than T1. However, considering the very low number of available data (few tens in the 5 months considered) and the fact that BGC-Argo floats data may exhibit wide oscillations over subsequent profiles (as shown in Fig. 10), the differences of skill performance statistics from one day of forecast to another might be considered cautionary.*

Finally, we have also stressed in the Conclusions (RP20,L12-14) that robust statistics require much longer time series of data and a larger number of BGC-Argo floats, which is becoming an urgent request for the observing systems to be used in operational biogeochemical oceanography (for both validation and assimilation purposes):

*Robust statistics require much longer time series of data and a larger number of BGC-Argo floats, which is becoming an urgent request for the observing systems to be used in operational biogeochemical oceanography (for both validation and assimilation purposes).*

Reviewer #2 – Technical comments

R2.a)   P4 L16-17: "are used respectively . . .module". I think you should switch both: "as input for the BFM optical module and to solve the gas air-sea exchanges"
We agree, text has been corrected.

R2.b)   P4 L21: temperature AND salinity
We agree, we have also included a comma before "and along-track".

R2.c)   P6 L14: the same AS
Corrected.

R2.d)   P7 L2: SchmeCHtig
Corrected.

R2.e)   P7 L16: relies instead of relays
Corrected.

R2.f)   P9 L6-7: surface bloomS in nwm appear (remove the s)
The sentence has been modified, according to R2.M2.

R2.g)   P9 L9-10: NODC-OGS instead of OGS-NODC for more consistency with the beginning of the manuscript
Corrected.

R2.h)   P10 L7: "a large seasonal cycles": remove the "s" in cycles
Corrected.

R2.i)   P11 L10 "river plume" : remove the "s"
Corrected.

R2.j)   P18 L1: represent a (remove the 'n')
Corrected.

R2.k)   Fig 14: in caption, 'increase' instead of 'increment'?
Yes, we modified with "increase".

**References**

[revised manuscript text omitted]

---

## Author Response (AR2)

Dear Editor,

please find here the revision of our manuscript following the technical corrections suggested by Referee #2. Referee's points are in *BLACK ITALIC*, our responses in BLUE and the modifications to the manuscript in *GREEN ITALIC*, with page/line numbers of the revised version. All the points were positively acknowledged.

In the text here submitted, we also corrected a small inaccuracy in Section 2.2 concerning the frequency of the data for the river nutrient boundary conditions (P6 L5) and some typos/wording through the text, also updating few references. Acknowledgements to Reviewers and a colleague from CINECA super-computing center have been added in the related section.

For sake of clarity, we uploaded the .docx file in track-changes mode. The uploaded pdf file is with all the changes accepted.

*Corrections:*

*R2.1: I would just add the reference of the product in the paragraph 3 "Reference datasets for validation" (OCEANCOLOUR_MED_CHL_L3_NRT_OBSERVATIONS_009_040) so as there is no ambiguity for the reader that it's the same product used for assimilation and validation.*

We added the reference product code at P7 L2 as already done in Section 2.2, where the same product is described for the first time. Please note that, for sake of readability, all the CMEMS product codes are added as footnotes in the text and the full acknowledgements are in the Reference section.

*R2.4: Maybe precise the year: 2016*

OK, now the sentence (at P5 L27) is:

*A spin-up period of 1 year (2016) repeated for 5 times in perpetual mode is carried out before the start of the simulation.*

*P3 L9: properties*

OK, corrected.

*P14 L11: I would have said product quality control procedure*

OK, corrected.

*P14 L12 : sensorS*

OK, corrected.

*P18 L27: in-depth*

OK, corrected.